# Prospective identification of functionally distinct stem cells and neurosphere-initiating cells in adult mouse forebrain

**John K Mich[1†], Robert AJ Signer[1†], Daisuke Nakada[2], André Pineda[1], Rebecca J Burgess[1], Tou Yia Vue[3], Jane E Johnson[3], Sean J Morrison[1]\***

[1]Department of Pediatrics, Children's Research Institute, Howard Hughes Medical Institute, University of Texas Southwestern Medical Center, Dallas, United States; [2]Department of Molecular and Human Genetics, Baylor College of Medicine, Houston, United States; [3]Department of Neuroscience, University of Texas Southwestern Medical Center, Dallas, United States

**Abstract** Neurosphere formation is commonly used as a surrogate for neural stem cell (NSC) function but the relationship between neurosphere-initiating cells (NICs) and NSCs remains unclear. We prospectively identified, and isolated by flow cytometry, adult mouse lateral ventricle subventricular zone (SVZ) NICs as Glast$^{mid}$EGFR$^{high}$PlexinB2$^{high}$CD24$^{−/low}$O4/PSA-NCAM$^{−/low}$Ter119/CD45$^−$ (GEPCOT) cells. They were highly mitotic and short-lived in vivo based on fate-mapping with *Ascl1*$^{CreERT2}$ and *Dlx1*$^{CreERT2}$. In contrast, pre-GEPCOT cells were quiescent, expressed higher Glast, and lower EGFR and PlexinB2. Pre-GEPCOT cells could not form neurospheres but expressed the stem cell markers *Slc1a3-CreER*$^T$, *GFAP-CreER*$^{T2}$, *Sox2*$^{CreERT2}$, and *Gli1*$^{CreERT2}$ and were long-lived in vivo. While GEPCOT NICs were ablated by temozolomide, pre-GEPCOT cells survived and repopulated the SVZ. Conditional deletion of the *Bmi-1* polycomb protein depleted pre-GEPCOT and GEPCOT cells, though pre-GEPCOT cells were more dependent upon Bmi-1 for *Cdkn2a* (*p16*$^{Ink4a}$) repression. Our data distinguish quiescent NSCs from NICs and make it possible to study their properties in vivo.

\*For correspondence: sean.morrison@utsouthwestern.edu

†These authors contributed equally to this work

## Introduction

Neural stem cells (NSCs) reside in two regions of the adult mammalian forebrain: the subgranular zone in the dentate gyrus and the subventricular zone in the lateral wall of the lateral ventricle (SVZ). SVZ NSCs persist throughout adult life (*Maslov et al., 2004*; *Molofsky et al., 2006*; *Imayoshi et al., 2008*), giving rise primarily to neurons in the olfactory bulb as well as some astrocytes in the olfactory bulb (*Lois and Alvarez-Buylla, 1994*; *Lois et al., 1996*; *Doetsch et al., 1999b*; *Ahn and Joyner, 2005*; *Kuo et al., 2006*; *Lagace et al., 2007*; *Merkle et al., 2007*; *Imayoshi et al., 2008*; *Chen et al., 2009*) and oligodendrocytes in the corpus callosum and cortex (*Nait-Oumesmar et al., 1999*; *Menn et al., 2006*). In vivo, these NSCs are quiescent (*Doetsch et al., 1999a*; *Pastrana et al., 2009*), resistant to anti-mitotic agents (*Morshead et al., 1994*; *Doetsch et al., 1999b, 2002*; *Giachino and Taylor, 2009*), long-lived (*Ahn and Joyner, 2005*; *Imayoshi et al., 2008*), and capable of regenerating the SVZ after injury (*Doetsch et al., 1999b, 2002*; *López-Juárez et al., 2013*).

Clonal colony-forming assays have been widely used to study neural stem/progenitor cells that give rise to neurons, astrocytes, and oligodendrocytes in adherent cultures (*Davis and Temple, 1994*) and non-adherent neurosphere cultures (*Reynolds and Weiss, 1992*). However, in the adult forebrain it remains uncertain whether these colonies are formed by quiescent neural stem cells (qNSCs) or by mitotically active and shorter-lived multipotent progenitors (*Morshead et al., 1994*; *Doetsch et al., 2002*;

**eLife digest** Neurons that arise in the adult nervous system originate from neural stem cells and neural progenitor cells. Neural stem cells have long lives, much of which they spend in a quiescent state. Neural stem cells can also give rise to neural progenitor cells, which proliferate rapidly during their short lives and then 'differentiate' into neurons or glia.

Unlike some other tissues, it has not been possible to identify or purify neural stem cells directly from the tissue. Consequently, neural stem and progenitor cells have usually been studied retrospectively, based on their ability to form colonies in laboratory cell cultures.

A region of the brain called the subventricular zone contains both neural stem cells and neural progenitor cells, and is one of only two regions of the brain where neural stem cells are found in adult mammals. When cells from the subventricular zone are cultured in a way that allows the cells to freely float around (rather than growing on a surface), a few percent form spherical colonies called neurospheres. Since neurosphere-forming cells can self-renew and differentiate into neurons and glia, the ability of cells to form neurospheres has generally been taken as evidence that they are stem cells. However, the exact relationship between neural stem cells and neurosphere-forming cells has been uncertain.

Now, Mich, Signer et al. have used a technique called flow cytometry to identify and isolate neural stem cells and neurosphere-forming cells directly from the subventricular zone. The neural stem cells, which Mich, Signer et al. term pre-GEPCOT cells (based on an acronym of the markers used to isolate the cells), were long-lived and quiescent, but they lacked the ability to form colonies in culture. The neurosphere-forming cells, named GEPCOT cells, were short-lived and highly proliferative in the brain. These results demonstrate that the cells that form neurospheres in culture are not stem cells at all, and that real stem cells are not able to form colonies under existing culture conditions.

The identification of undifferentiated pre-GEPCOT and GEPCOT cells will make it possible to directly study the properties of these cells inside the mouse brain, and to isolate live cells to test how they function. The results also highlight the need for new tests to study neural stem cell function, given that current tests using neurospheres do not detect stem cells as commonly assumed.

*Reynolds and Rietze, 2005*; *Pastrana et al., 2009*, *2011*). Moreover, reliance upon retrospective colony-formation assays makes it impossible to directly study qNSCs or NICs as they exist in vivo.

Pioneering work by Alvarez-Buylla and colleagues has demonstrated the existence of a lineage of NSCs and transit amplifying cells in the SVZ that gives rise to neuronal progenitors throughout life (*Lois and Alvarez-Buylla, 1994*; *Doetsch et al., 1997*, *1999a*; *Mirzadeh et al., 2008*; *Ihrie and Alvarez-Buylla, 2011*). By electron microscopy and immunofluorescence analysis they identified GFAP-expressing type B cells that appear to be the qNSCs. These cells are resistant to anti-mitotic agents such as AraC and appear capable of repopulating the SVZ after AraC treatment (*Doetsch et al., 1999a*, *1999b*, *2002*). Type B cells are thought to give rise to *Ascl1*- and *Dlx2*-expressing type C cells, which are mitotically active and ablated by anti-mitotic agents (*Morshead et al., 1994*; *Doetsch et al., 1999b*, *2002*; *Pastrana et al., 2009*). The type C cells give rise to Dcx- and PSA-NCAM-expressing type A neuronal progenitors. These studies have provided a critical framework for understanding the SVZ neurogenic lineage, though the inability to purify live cells from each stage of this hierarchy has hampered efforts to assess their properties.

The inability to prospectively identify and isolate uncultured stem cells from the central nervous system (CNS) has contributed to uncertainty regarding the relationship between qNSCs and NICs. SVZ NICs have been enriched by flow cytometry based on CD15 expression (*Capela and Temple, 2002*) or ROS levels (*Le Belle et al., 2011*). Multiple lines of evidence have demonstrated that NSCs express GFAP and that these cells sustainably contribute to neurogenesis in vivo (*Doetsch et al., 1997*, *1999a*, *1999b*, *2002*; *Imura et al., 2003*; *Morshead et al., 2003*; *Garcia et al., 2004*; *Mirzadeh et al., 2008*; *Pastrana et al., 2009*; *Beckervordersandforth et al., 2010*; *Giachino et al., 2013*). Pastrana et al. identified quiescent GFAP-GFP$^+$EGFR$^-$ SVZ cells and speculated that these cells include NSCs that give rise to more mitotically active GFAP-GFP$^+$EGFR$^+$ and GFAP-GFP$^-$EGFR$^+$ NICs but did not test this

by fate-mapping (*Pastrana et al., 2009*). In contrast, it has been suggested that nearly all GFAP-GFP⁺CD133⁺ cells are NICs and that these cells are the qNSCs in the SVZ (*Beckervordersandforth et al., 2010*). Efforts toward prospective identification have therefore generated conflicting results about whether NICs are quiescent or mitotically active in vivo and regarding their relationship to NSCs.

We have identified regulators of CNS stem cell self-renewal based on their ability to regulate SVZ proliferation and neurogenesis in vivo as well as multipotent NIC self-renewal in culture (*Molofsky et al., 2003*, *2006*; *Nishino et al., 2008*; *Chuikov et al., 2010*). However, an important question that we have not been able to address directly is whether those genes are necessary for NSC maintenance in vivo. Impaired NIC self-renewal in culture may not reflect reduced NSC self-renewal in vivo (*Joseph and Morrison, 2005*; *He et al., 2009*). For example, the polycomb transcriptional repressor *Bmi-1* is thought to be required for NSC self-renewal (*Molofsky et al., 2003*; *Bruggeman et al., 2005*; *Molofsky et al., 2005*; *Zencak et al., 2005*; *Bruggeman et al., 2007*; *Fasano et al., 2009*). However, these studies were performed in germline knockout mice that generally die within a month after birth (*van der Lugt et al., 1994*; *Jacobs et al., 1999*; *Lessard and Sauvageau, 2003*; *Park et al., 2003*). Thus, it has not been possible to test whether Bmi-1 is autonomously required by NSCs in the adult brain or whether NSCs differ from NICs in their dependence upon Bmi-1.

Here we report the prospective identification of two phenotypically and functionally distinct populations of cells in the SVZ: GEPCOT cells and pre-GEPCOT cells. The pre-GEPCOTs accounted for 6 ± 3% of adult mouse SVZ cells, were highly quiescent, lacked the ability to form neurospheres or adherent colonies in culture, and included type B1 cells based on marker expression, morphology, and position in vivo. These cells contained long-lived qNSCs based on both fate mapping and temozolomide resistance. GEPCOTs were distinguished by lower GFAP and Glast expression and higher EGFR and PlexinB2 expression. These cells accounted for 3.2 ± 0.7% of cells in the adult mouse SVZ, were highly mitotically active, highly enriched for NICs, and included type C cells based on marker expression, morphology, and position in vivo. Based on fate-mapping these cells were short-lived in the SVZ. Our data thus provide methods to prospectively identify and distinguish qNSCs from NICs.

## Results

### Prospective identification of NICs

We enzymatically dissociated adult mouse SVZ cells then sorted cells by flow cytometry into non-adherent cultures at clonal density (0.66 cells/µl of culture medium). We always replated neurospheres to adherent secondary cultures to assess differentiation into TuJ1⁺ neurons, GFAP⁺ astrocytes, and O4⁺ oligodendrocytes. On average, 1.8 ± 0.4% of SVZ cells formed neurospheres (>50 µm diameter) and 75% of those neurospheres underwent multilineage differentiation (1.4 ± 0.3% of SVZ cells).

We systematically screened 383 antibodies against 330 distinct cell surface antigens (*Supplementary file 1A*) to identify markers that could enrich NICs (*Figure 1A*). We identified 49 markers by flow cytometry that were heterogeneously expressed among dissociated SVZ cells. For each of these markers we sorted SVZ cells that differed in their level of staining into non-adherent cultures and assessed neurosphere formation. We found 17 markers that enriched NICs relative to unfractionated SVZ cells (*Supplementary file 1A*). We multiplexed combinations of these markers to optimize enrichment while ensuring that most NICs were retained within the sorted population.

We greatly enriched NICs by isolating live (4′,6-diamidino-2-phenylindole (DAPI) negative) SVZ cells that expressed moderate levels of Glast, high levels of Epidermal Growth Factor Receptor (EGFR), high levels of PlexinB2, negative to low levels of CD24, negative to low levels of O4 and PSA-NCAM, and were negative for the hematopoietic markers Ter119 and CD45. We refer to these Glast^mid^EGFR^high^PlexinB2^high^CD24^−/low^O4/PSA-NCAM^−/low^Ter119/CD45⁻ cells as GEPCOT cells (*Figure 1B*, *Figure 1—figure supplement 1A,B*). GEPCOTs accounted for 3.2 ± 0.7% of all SVZ cells (*Figure 1B*). On average, 36 ± 6% of GEPCOT cells formed neurospheres (>50 µm diameter) and 74% of those neurospheres underwent multilineage differentiation (*Figure 1C*). Nearly all of the neurospheres (91%) could be passaged (data not shown). On average, each neurosphere gave rise to 53 ± 41 multipotent secondary neurospheres upon dissociation and replating, demonstrating self-renewal potential. Most NICs from the SVZ were contained within this GEPCOT population (*Figure 1—figure supplement 1C*). Given that individual NICs are unlikely to form colonies with 100% efficiency after dissociation and flow cytometry, most GEPCOT cells likely have the potential to form neurospheres.

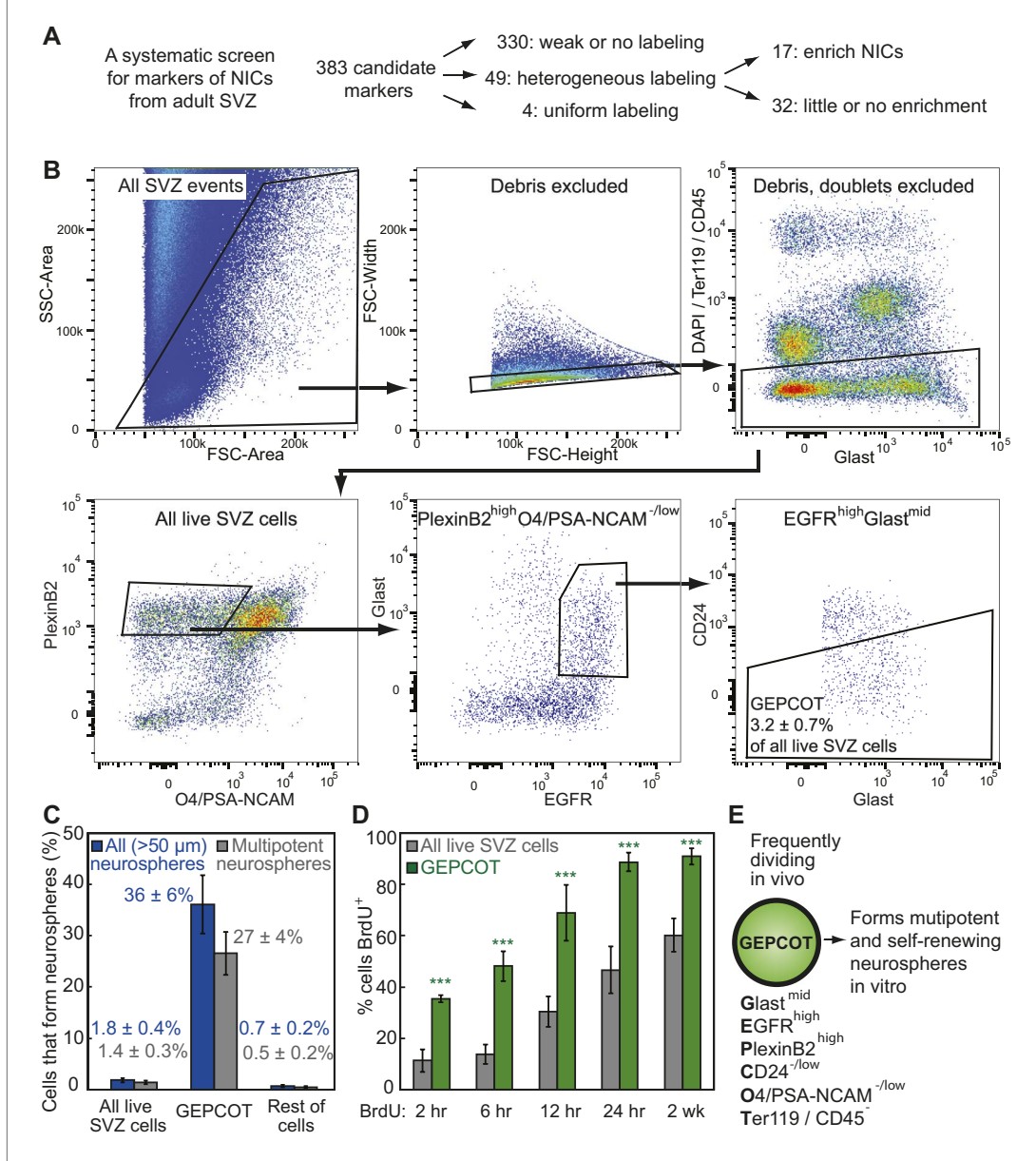

**Figure 1**. Prospective identification and isolation of neurosphere-initiating cells. (**A**) A screen of antibodies identified cell surface markers of NICs in the adult mouse SVZ. (**B**) Flow cytometric gating strategy to isolate GEPCOT cells (**G**last$^{mid}$**E**gfr$^{high}$**P**lexinB2$^{high}$**C**D24$^{-/low}$**O**4/PSA-NCAM$^{-low}$**T**er119/CD45$^{-}$) that represent 3.2 ± 0.7% of young adult mouse SVZ cells. Plots represent one representative experiment from at least nine independent experiments. For more information on gating see *Figure 1—figure supplement 1*. (**C**) Frequency of all neurospheres (>50 μm) and multipotent neurospheres formed by unfractionated SVZ cells (DAPI/CD45/Ter119$^-$), GEPCOT cells, and remaining live SVZ cells outside of the GEPCOT population (n = 9 independent experiments). (**D**) Frequency of BrdU$^+$ SVZ cells or GEPCOT cells after BrdU pulses in vivo (n = 5–11 mice/time point in 2–3 independent experiments). (**E**) GEPCOT cells efficiently formed multipotent neurospheres in vitro and were mitotically active in vivo. Data always represent mean ± SD. Statistical significance was assessed with two-tailed *t* tests, ***p<0.001.

The following figure supplements are available for figure 1:

**Figure supplement 1**. Isolating GEPCOT cells by flow cytometry.

## NICs are highly proliferative and short-lived in vivo

The ability to prospectively identify NICs made it possible to assess their cell cycle distribution in vivo by administering bromodeoxyuridine (BrdU) to mice. After just a 2 hour pulse of BrdU, 35 ± 2% of

GEPCOTs were already BrdU$^+$ as compared to only 11 ± 4% of unfractionated SVZ cells (*Figure 1D*). Longer pulses of BrdU progressively increased the labeling of the GEPCOT cells. After a 24-hr pulse of BrdU, 89 ± 4% of GEPCOTs were BrdU$^+$ as compared to 47 ± 9% of unfractionated SVZ cells (*Figure 1D*). GEPCOTs are thus highly mitotically active and enriched for dividing cells relative to unfractionated SVZ cells. The observation that nearly all GEPCOTs incorporated BrdU within 24 hr indicates that few, if any, GEPCOTs are quiescent and that NICs are highly proliferative in vivo.

To assess the persistence of NICs in vivo we performed a series of lineage tracing experiments. Guided by previous observations (*Doetsch et al., 2002*; *Ahn and Joyner, 2005*; *Arnold et al., 2011*; *Kim et al., 2011*; *Taniguchi et al., 2011*; *Lee et al., 2012*; *Wang et al., 2012*) we screened candidate CreER$^{T2}$ alleles for the ability to recombine a conditional *Rosa26-loxp-tdTomato* reporter within NICs in vivo after 5 consecutive days of tamoxifen injection (*Figure 2A*; 80 mg/kg body mass/day i.p.). We found that *Dlx1$^{CreERT2}$* (*Taniguchi et al., 2011*), *Ascl1$^{CreERT2}$* (*Kim et al., 2011*), *Gli1$^{CreERT2}$* (*Ahn and Joyner, 2005*), *Slc1a3-CreER$^T$* (*Wang et al., 2012*), and *Sox2$^{CreERT2}$* (*Arnold et al., 2011*) labeled 11–94% of NICs 2 days after tamoxifen administration (*Figure 2B–F*). With each Cre allele and at each time point, the frequency of tdTomato$^+$ GEPCOTs was statistically indistinguishable from the frequency of tdTomato$^+$ neurospheres that arose in culture from unfractionated SVZ cells (*Figure 2B–F*). This independently confirms that GEPCOT markers reliably identify uncultured cells with the ability to form neurospheres.

To assess the persistence of NICs in vivo we quantified the frequencies of all SVZ cells, GEPCOTs, and neurospheres that were tdTomato$^+$ at 2, 7, 28, or 60 days after tamoxifen treatment. In the absence of tamoxifen we detected no tdTomato expression in SVZ cells, GEPCOTs, or cultured neurospheres in *Dlx1$^{CreERT2}$* mice (*Figure 2B*). 2 days after tamoxifen treatment, *Dlx1$^{CreERT2}$* labeled 51 ± 6% of SVZ cells, 15 ± 6% of GEPCOTs, and 11 ± 4% of neurospheres (*Figure 2B*). 7 days after tamoxifen this significantly (p<0.001) declined to 8 ± 5% of SVZ cells, 0.7 ± 0.9% of GEPCOTs, and 0.9 ± 1% of neurospheres (*Figure 2B*). 28 days after tamoxifen only rare SVZ cells, GEPCOTs, and neurospheres were labeled (*Figure 2B*). *Dlx1$^{CreERT2}$*-expressing NICs thus persist in vivo for less than 7 days.

*Ascl1$^{CreERT2}$* labeled most GEPCOTs and neurospheres. In the absence of tamoxifen, only 0.02% of SVZ cells, and no GEPCOTs or neurospheres, were tdTomato$^+$ (*Figure 2C*). 2 days after tamoxifen, *Ascl1$^{CreERT2}$* labeled 34 ± 7% of SVZ cells, 63 ± 8% of GEPCOTs, and 64 ± 6% of neurospheres (*Figure 2C*). 7 days after tamoxifen a similar fraction of SVZ cells remained tdTomato$^+$ (35 ± 3%) but the fraction of labeled GEPCOTs and neurospheres significantly (p<0.01) declined to 40 ± 4% and 45 ± 3%, respectively (*Figure 2C*). 28 days after tamoxifen only 16 ± 0.9% of SVZ cells, 16 ± 0.6% of GEPCOTs, and 19 ± 1.0% of neurospheres were labeled. This further decreased to 7.5 ± 1.2% of SVZ cells, 12 ± 0.8% of GEPCOTs, and 8 ± 3% of neurospheres at 60 days after tamoxifen (*Figure 2C*). This demonstrates that most *Ascl1$^{CreERT2}$*-expressing NICs persist in vivo for less than 28 days.

In contrast to cells labeled with *Dlx1$^{CreERT2}$* or *Ascl1$^{CreERT2}$*, cells labeled with *Gli1$^{CreERT2}$*, *Slc1a3-CreER$^T$*, or *Sox2$^{CreERT2}$* exhibited a sustained contribution to the pools of SVZ cells, GEPCOTs, and NICs for at least 60 days after tamoxifen administration (*Figure 2D–F*). This suggested the existence of a long-lived NSC population in the SVZ that replenishes NICs and is marked by *Gli1$^{CreERT2}$*, *Slc1a3-CreER$^T$*, and *Sox2$^{CreERT2}$* expression (*Figure 2G*).

## Identification of GFAP-expressing NSCs in vivo

To test whether there is an earlier stem cell population in the SVZ we sought a CreER$^{T2}$ allele that recombines in long-lived NSCs but not in NICs. Since prior studies have implicated GFAP as a marker of type B cells and NSCs in the adult SVZ (*Doetsch et al., 1997*, *1999a*, *2002*; *Mirzadeh et al., 2008*; *Giachino and Taylor, 2009*; *Giachino et al., 2013*) we performed lineage-tracing experiments with *GFAP-CreER$^{T2}$* (*Hirrlinger et al., 2006*). In the absence of tamoxifen, only 0.02% of SVZ cells, and no GEPCOTs or neurospheres, were tdTomato$^+$ (*Figure 3A*). 2 days after tamoxifen treatment, *GFAP-CreER$^{T2}$* labeled only 5 ± 0.6% of SVZ cells, 7 ± 2% of GEPCOTs, and 8 ± 0.9% of neurospheres (*Figure 3A*). 7 days after tamoxifen this increased significantly (p<0.05) to 11 ± 4.3% of SVZ cells, 19 ± 5.5% of GEPCOTs, and 18 ± 6.1% of neurospheres (*Figure 3A*). By 28 days after tamoxifen this further increased to 26 ± 3.7% of SVZ cells, 32 ± 4.7% of GEPCOTs, and 35 ± 7.2% of neurospheres (*Figure 3A*). At 60 days after tamoxifen we observed 23 ± 2.6% of SVZ cells, 34 ± 8.0% of GEPCOTs, and 37 ± 4.2% of neurospheres labeled (*Figure 3A*). *GFAP-CreER$^{T2}$*-expressing NSCs thus make a sustained and growing contribution to the SVZ, in contrast to NICs labeled by *Dlx1$^{CreERT2}$* or *Ascl1$^{CreERT2}$*.

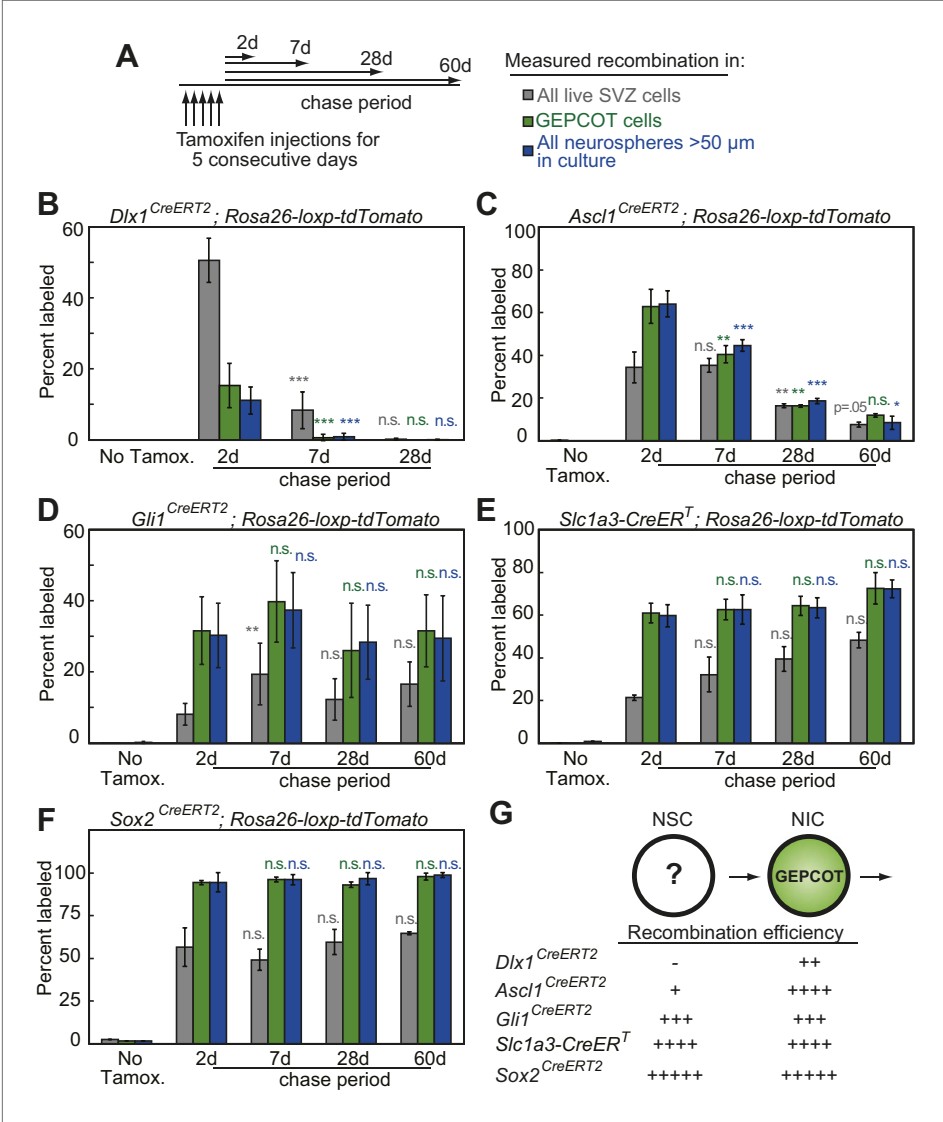

**Figure 2**. Individual NICs persist only transiently within the SVZ in vivo but are constantly replenished by more primitive neural stem cells. (**A**) Mice bearing inducible Cre alleles and the *Rosa26-loxp-tdTomato* conditional reporter were induced with 5 consecutive days of tamoxifen injections (80 mg/kg/day) then chased for 2 to 60 days before analysis of the percentage of labeled SVZ cells or GEPCOT cells in vivo or the percentage of labeled neurospheres (>50 μm) in culture. (**B–F**) Cells marked by recombination of *Dlx1^CreERT2* (**B**, n = 4–7 mice/time point in four independent experiments), *Ascl1^CreERT2* (**C**, n = 3 or 4 mice/time point in three independent experiments), *Gli1^CreERT2* (**D**), *Slc1a3-CreER^T* (**E**), or *Sox2^CreERT2* (**F**, n = 3–4 mice/time point in four independent experiments for **D–F**). (**G**) A model consistent with the fate mapping data involving a NSC population that gives rise to a transient NIC population. Data represent mean ± SD. Statistical significance was tested among sequential days of analysis (7d was compared to 2d, 28d was compared to 7d, and 60d was compared to 28d) with a one-way ANOVA followed by Tukey's post-hoc tests for multiple comparisons, *p<0.05, **p<0.01, ***p<0.001, n.s. not significant.

To better characterize the *GFAP-CreER^T2*-expressing SVZ cells we examined the surface markers expressed by tdTomato⁺ cells 2 days after tamoxifen treatment of *GFAP-CreER^T2*; *Rosa26-loxp-tdTomato* mice. By flow cytometry, tdTomato⁺ cells were highly enriched for Glast^high EGFR^−/low PlexinB-2^mid CD24^−/low O4/PSA-NCAM^−/low Ter119/CD45⁻ (pre-GEPCOT) cells (*Figure 3B–D*). These cells differed from GEPCOT cells in that they exhibited higher levels of Glast and lower levels of EGFR and PlexinB2 expression. pre-GEPCOT cells made up only 6 ± 3% of all SVZ cells (*Figure 3E*) and yet, based on *GFAP-CreER^T2* lineage tracing, gave rise to many SVZ cells, GEPCOTs, and NICs within 28 days of tamoxifen

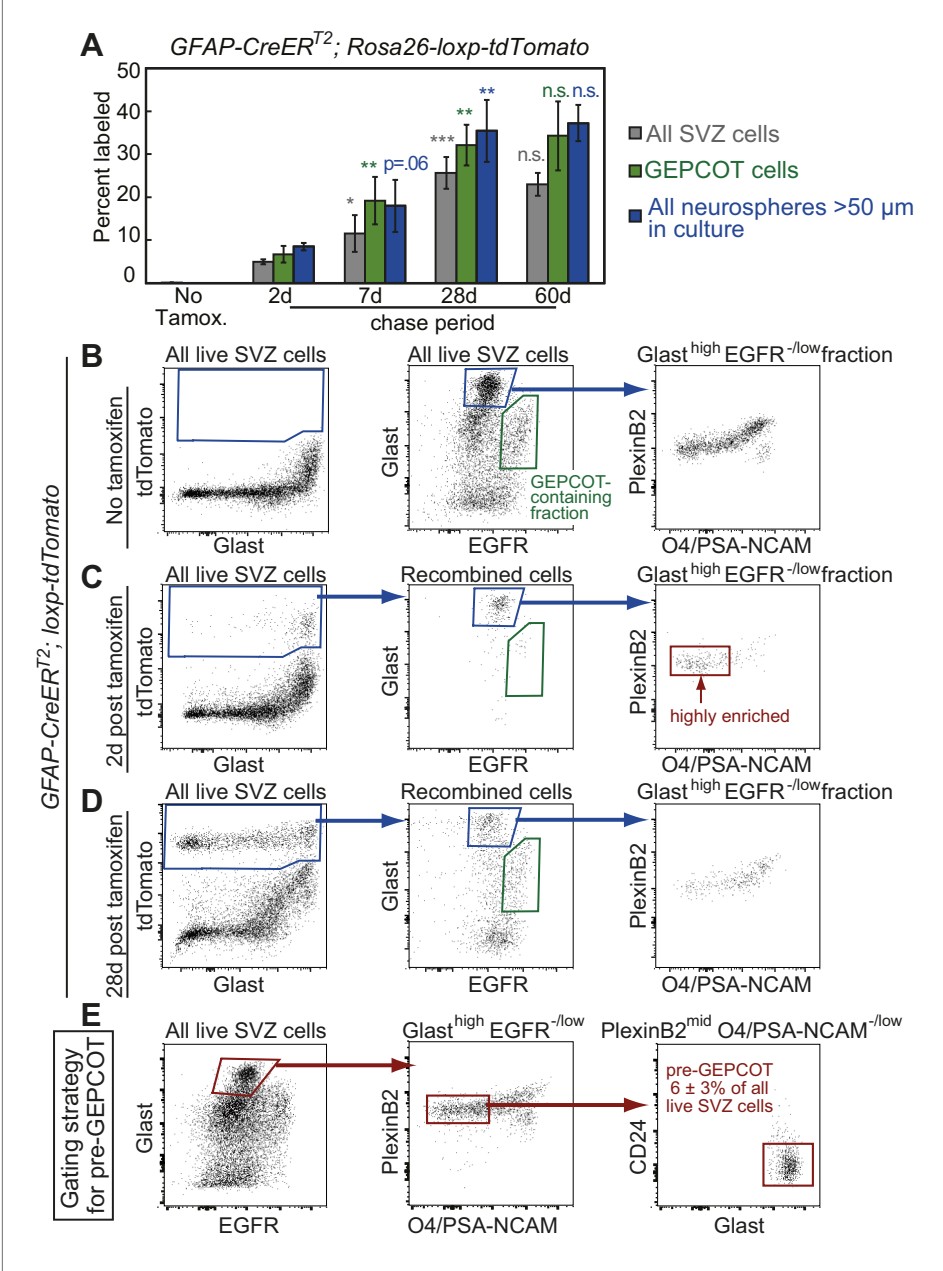

**Figure 3**. Identification of a pre-GEPCOT population that gives rise to GEPCOT NICs in vivo. (**A**) *GFAP-CreER*[T2]; *Rosa26-loxp-tdTomato* mice were induced with tamoxifen for 5 days, chased for 2 to 60 days without tamoxifen, then conditional reporter expression was analyzed in all SVZ cells, GEPCOTs, and cultured neurospheres (n = 3–5 mice/time point in four independent experiments). (**B**–**D**) *GFAP-CreER*[T2]; *Rosa26-loxp-tdTomato* mice were analyzed by flow cytometry to quantify reporter expression in SVZ cells without tamoxifen (**B**), at 2 days (**C**) or 28 days (**D**) after tamoxifen treatment. (**B**) Without tamoxifen, virtually no SVZ cells expressed the reporter. For comparison to **C** and **D**, control SVZ cells are shown stained for Glast and EGFR as well as the distribution of the Glast[high]EGFR[−/low] fraction with respect to PlexinB2 and O4/PSA-NCAM staining. (**C**) 2 days after tamoxifen, the tdTomato[+] cells labeled by recombination by the *GFAP-CreER*[T2] stem cell marker were Glast[high]Egfr[−/low]PlexinB2[mid]CD24[−/low]O4/ PSA-NCAM[−/low]Ter119/Cd45[−] (CD24 expression is not shown here), which we describe as 'pre-GEPCOT' cells. (**D**) By 28 days after tamoxifen, Glast[mid]Egfr[high]PlexinB2[high]CD24[−/low]O4/PSA-NCAM[−/low]Ter119/CD45[−] GEPCOT cells were also labeled. (**E**) The gating strategy to identify the GFAP-expressing pre-GEPCOT population that comprises 6 ± 3% of SVZ cells. Statistical significance was tested among sequential days of analysis with a one-way ANOVA followed by Tukey's post-hoc tests for multiple comparisons. *p<0.05, **p<0.01, ***p<0.001, n.s. not significant.

treatment (*Figure 3A*). The *GFAP-CreER^T2*-expressing pre-GEPCOTs therefore included NSCs with a much greater capacity to contribute to the SVZ than *Ascl1^CreERT2*-expressing NICs.

To confirm this we sorted pre-GEPCOT cells to test whether they stain with an anti-GFAP antibody. GFAP expression was heterogeneous within the pre-GEPCOT population, with 21 ± 7% being GFAP^high and 33 ± 2% being GFAP^low (*Figure 4A*). This suggests that the pre-GEPCOT population is much more highly enriched for NSCs as compared to unfractionated SVZ cells, which contained only 2 ± 1% GFAP^high cells and 6 ± 3% GFAP^low cells (both p<0.01 relative to pre-GEPCOT cells; *Figure 4A*).

## pre-GEPCOT cells are long-lived qNSCs

Whereas GEPCOTs were highly enriched for multipotent NICs (*Figure 1C*), pre-GEPCOTs were unable to form neurospheres. We sorted unfractionated SVZ cells into culture and found that 1.9 ± 0.5% of the cells formed neurospheres and 2.0 ± 0.4% formed adherent colonies (*Figure 4B*). Under the same conditions 32 ± 7% of GEPCOTs formed neurospheres and 38 ± 13% formed adherent colonies (*Figure 4B*). In contrast, only very rare pre-GEPCOT cells formed neurospheres (1/1663 cells) and only 0.3 ± 0.4% (9/1973 cells) formed adherent colonies (*Figure 4B*).

To assess whether pre-GEPCOT cells are quiescent or frequently dividing in vivo, we administered pulses of BrdU. Although 47 ± 9% of SVZ cells and 89 ± 4% of GEPCOT cells (*Figure 1D*) incorporated a 24-hr pulse of BrdU in these experiments, only 3 ± 3% of pre-GEPCOT cells incorporated BrdU in the same mice (*Figure 4C*). Similarly, 60 ± 6% of SVZ cells and 91 ± 3% of GEPCOT cells (*Figure 1D*) incorporated a 2-week pulse of BrdU in these experiments, but only 4 ± 2% of pre-GEPCOT cells incorporated BrdU (*Figure 4C*). These data indicate that pre-GEPCOT cells are quiescent in the SVZ.

To assess whether fate-mapping with CreER^T2 alleles is consistent with the existence of a pre-GEPCOT qNSC population we assessed whether they recombined in pre-GEPCOT cells. *Dlx1^CreERT2*, which labeled 51 ± 6% of all SVZ cells and 15 ± 6% of GEPCOTs at 2 days after tamoxifen treatment (*Figure 2B*), did not label any pre-GEPCOT cells at 2, 7, or 28 days after tamoxifen treatment (*Figure 4D*). *Ascl1^CreERT2*, which labeled 34 ± 7% of all SVZ cells and 63 ± 8% of GEPCOTs at 2 days after tamoxifen (*Figure 2C*), labeled fewer than 1% of pre-GEPCOT cells at 2, 7, 28, and 60 days after tamoxifen (*Figure 4E*). Thus, both of the CreER^T2 alleles that exhibited declining contributions to the SVZ upon fate-mapping labeled GEPCOTs but not pre-GEPCOTs. Neither the *Dlx1^CreERT2*-expressing GEPCOT NICs nor the *Ascl1^CreERT2*-expressing GEPCOT NICs gave rise to significant numbers of pre-GEPCOT qNSCs in vivo, at least under steady-state conditions.

To assess whether pre-GEPCOTs contain qNSCs we tested whether the CreER^T2 alleles that sustainably contributed to the SVZ over time (*Gli1^CreERT2*, *Slc1a3-CreER^T*, and *Sox2^CreERT2*) labeled pre-GEPCOT cells. *Gli1^CreERT2*, which sustainably labeled 8–19% of all SVZ cells and 26–40% of GEPCOTs for 60 days after tamoxifen treatment (*Figure 2D*), also sustainably labeled 7–13% of pre-GEPCOT cells at 2 to 60 days after tamoxifen treatment (*Figure 4F*). *Slc1a3-CreER^T*, which sustainably labeled 21–48% of all SVZ cells and 61–72% of GEPCOTs for 60 days after tamoxifen (*Figure 2E*), also labeled 31–36% of pre-GEPCOT cells 2 to 60 days after tamoxifen (*Figure 4G*). *Sox2^CreERT2*, which sustainably labeled 49–65% of all SVZ cells and 92–97% of GEPCOTs for 60 days after tamoxifen (*Figure 2F*), also sustainably labeled 87–93% of pre-GEPCOT cells 2 to 60 days after tamoxifen (*Figure 4H*). Thus all of the CreER^T2 alleles that gave a sustained contribution to NIC and SVZ labeling also labeled pre-GEPCOTs, consistent with the suggestion that pre-GEPCOT cells contain qNSCs. Moreover, every CreER^T2 allele that labeled pre-GEPCOT cells exhibited sustained labeling of not only pre-GEPCOT cells but also GEPCOTs, NICs, and unfractionated SVZ cells.

We tested whether the *GFAP-CreER^T2* recombination pattern was consistent with pre-GEPCOT qNSCs giving rise to GEPCOT NICs in vivo. 2 days after tamoxifen administration to *GFAP-CreER^T2*; *Rosa26-loxp-tdTomato* mice, the tdTomato label was present mainly in pre-GEPCOT cells. On average, 29 ± 7% of pre-GEPCOT cells were tdTomato^+ whereas only 6.7 ± 1.9% of GEPCOT cells, 2.2 ± 0.5% of PSA-NCAM^highCD24^mid neuroblasts (*Pastrana et al., 2009*), and 3.0 ± 0.8% of other SVZ cells were labeled (*Figure 4I*). These data indicate that *GFAP-CreER^T2* recombines efficiently in pre-GEPCOTs but not GEPCOT NICs in vivo. 28 days after tamoxifen treatment, the percentage of labeled pre-GEPCOT cells increased to 44 ± 5.8% of cells (*Figure 4I*). The percentages of labeled GEPCOTs, neuroblasts, and other SVZ cells also increased significantly (p<0.01) by 28 days–32 ± 4.7%, 35 ± 6.8%, and 20 ± 2.2%, respectively. At 60 days after tamoxifen, we continued to observe strong labeling in pre-GEPCOTs, GEPCOTs, and neuroblasts (*Figure 4I*). Similar trends were apparent when the data

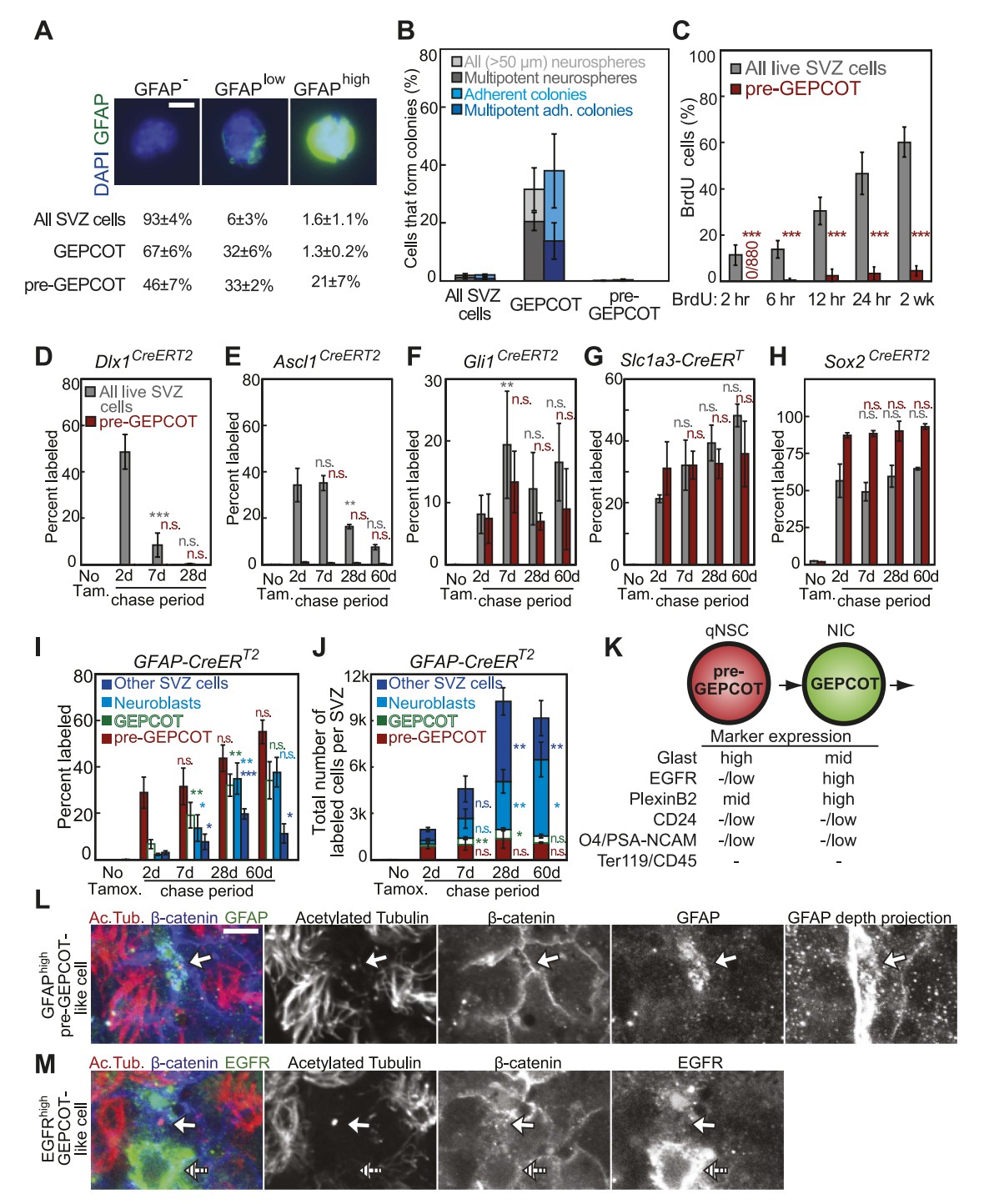

**Figure 4.** GFAP-expressing pre-GEPCOT cells are quiescent in vivo but make an enduring contribution to the SVZ. (**A**) Antibody staining for GFAP among SVZ cells, GEPCOTs, and pre-GEPCOTs (scale bar = 5 μm) (n = 3 mice from three independent experiments). (**B**) The frequencies of neurospheres >50 μm, multipotent neurospheres, adherent colonies, and multipotent adherent colonies (4–6 independent experiments each) formed by unfractionated live SVZ cells, GEPCOTs, and pre-GEPCOTs. (**C**) Frequency of BrdU⁺ SVZ cells or pre-GEPCOT cells after BrdU pulses in vivo (n = 5–11 mice/time point in 2–3 independent experiments; note that SVZ data are from *Figure 1D* for comparison purposes but were obtained in the same experiments). (**D–H**) Conditional reporter expression in pre-GEPCOT cells at varying times after recombination with the indicated Cre alleles. These data are from the same fate-mapping experiments as shown in *Figures 2 and 3*, including the same SVZ data for comparison purposes. Cre alleles that only transiently contributed to the SVZ (*Dlx1^CreERT2* and *Ascl1^CreERT2*) did not recombine in pre-GEPCOT cells, while Cre alleles that gave enduring contributions to the SVZ

*Figure 4. Continued*

(*Gli1*$^{CreERT2}$, *Slc1a3-CreER*$^T$, and *Sox2*$^{CreERT2}$) did recombine in pre-GEPCOT cells. (**I–J**) The frequencies and numbers of labeled pre-GEPCOT cells, GEPCOT cells, neuroblasts, and other SVZ cells at varying times after recombination by *GFAP-CreER*$^{T2}$ (n = 3–5 mice/time point in four independent experiments). (**K**) Markers that distinguish pre-GEPCOT from GEPCOT cells. (**L–M**) Whole-mount SVZs were stained with anti-acetylated tubulin (red), anti-β-catenin (blue), and either anti-GFAP (**L**, green) or anti-EGFR (**M**, green) antibodies and pinwheel structures were inspected for the presence of GFAP$^{high}$ pre-GEPCOT cells (open arrow in **L** and **M**) and EGFR$^{high}$ GEPCOT cells (hatched arrow in **M**) by confocal microscopy. Images were taken at the apical surface except the GFAP depth projection which is a composite of 11 images at 2 μm intervals into the tissue. Scale bar = 5 μm. All data represent mean ± SD. Statistical significance of differences between SVZ and pre-GEPCOT cells in **C** was assessed with two-tailed student's *t* tests. Statistical significance of differences in **D–J** (among time points) was tested with a one-way ANOVA followed by Tukey's post-hoc tests for multiple comparisons. *p<0.05, **p<0.01, ***p<0.001, n.s. not significant.

were expressed in absolute numbers (*Figure 4J*). These data suggest *GFAP-CreER*$^{T2}$-expressing pre-GEPCOT cells include qNSCs that give rise to NICs, neuroblasts, and other SVZ cells (*Figure 4K*).

We examined the localization of pre-GEPCOT and GEPCOT cells in whole mount stains of the SVZ. pre-GEPCOTs were distinguished by GFAP expression, a marker of type B cells (*Doetsch et al., 1999a*) and GEPCOTs were distinguished by EGFR expression, a marker of type C cells (*Doetsch et al., 2002*). We stained whole-mount SVZs with antibodies against acetylated tubulin and β-catenin to detect 'pinwheel' structures associated with type B1 stem cells in the SVZ (*Mirzadeh et al., 2008*) then also stained with antibodies against GFAP and EGFR. Consistent with prior results (*Mirzadeh et al., 2008*), we observed GFAP$^{high}$ cells at the center of the pinwheel structures with a single primary cilium on the apical surface contacting the ventricle and a long basal process (*Figure 4L*). Since many pre-GEPCOTs recombined with *GFAP-CreER*$^{T2}$ but virtually no GEPCOTs did (*Figure 4I*), these data indicate that many pre-GEPCOT cells have the morphology and position of type B1 cells in the SVZ. In contrast, EGFR$^{high}$ cells had type C cell morphology: round, unciliated, and generally not contacting the ventricle, consistent with *Doetsch et al. (2002)* (*Figure 4M*). Since all GEPCOT cells express high levels of EGFR (*Figure 1B*) while pre-GEPCOT cells have low levels of EGFR (*Figure 3D*), these data suggest that many GEPCOT cells have the morphology and position of type C cells.

## Anti-mitotic agent eliminates GEPCOT NICs but not pre-GEPCOT qNSCs

Our observation that pre-GEPCOT cells include qNSCs raised the question of whether pre-GEPCOT cells are resistant to anti-mitotic agents. To test this, we treated mice for 3 consecutive days with the CNS-penetrating DNA-alkylating agent temozolomide (TMZ; 100 mg/kg/day i.p.) (*Garthe et al., 2009*) then assessed SVZ proliferation, composition, and neurosphere formation 3 to 90 days later (*Figure 5A*). 3 days after TMZ, mice appeared healthy and had lost less than 3% of their body mass as compared to before TMZ treatment (data not shown). 3 days after TMZ treatment we observed a 23% reduction in the total number of cells per SVZ and this reduction remained nearly constant over the next 90 days (*Figure 5B*). 3 days after TMZ we observed an 80% reduction in the number of dividing SVZ cells, based on BrdU incorporation (*Figure 5C*, p<0.001). The number of dividing SVZ cells slowly increased over time, recovering to 50% of normal by 90 days after TMZ treatment (*Figure 5C*). 3 days after TMZ only 0.7% of the NICs observed in saline-treated control mice remained in TMZ-treated mice (*Figure 5D*, p<0.001). By 16 days after TMZ treatment NICs recovered to 20% of normal levels (*Figure 5D*, p<0.001). The numbers of NICs continued to increase over time, recovering to 55% of normal by 90 days after TMZ (*Figure 5D*, p<0.001). NICs are thus virtually completely eliminated by TMZ but regenerate over time as expected.

Consistent with the effects of TMZ on NICs, TMZ treatment also eliminated nearly all GEPCOT cells. 3 days after TMZ treatment, only 10% of the GEPCOTs observed in saline-treated control mice remained in TMZ treated mice (*Figure 5E*, p<0.001). By 16 days after TMZ treatment, GEPCOTs recovered to 32% of normal levels (*Figure 5E*, p<0.001). GEPCOTs recovered to 55% of normal levels by 90 days after TMZ treatment (*Figure 5E*, p<0.001).

The numbers of pre-GEPCOT cells in the SVZ were not affected by TMZ treatment (*Figure 5F*). If pre-GEPCOT qNSCs regenerate NICs after TMZ treatment then serial TMZ treatment might be expected to deplete pre-GEPCOT cells. To test this we administered two rounds of TMZ 12 days apart (*Figure 5G*). Under these circumstances, pre-GEPCOT cells were not depleted 3 days after the second round of TMZ treatment but they were depleted by 90 days after the second round of TMZ (*Figure 5L*).

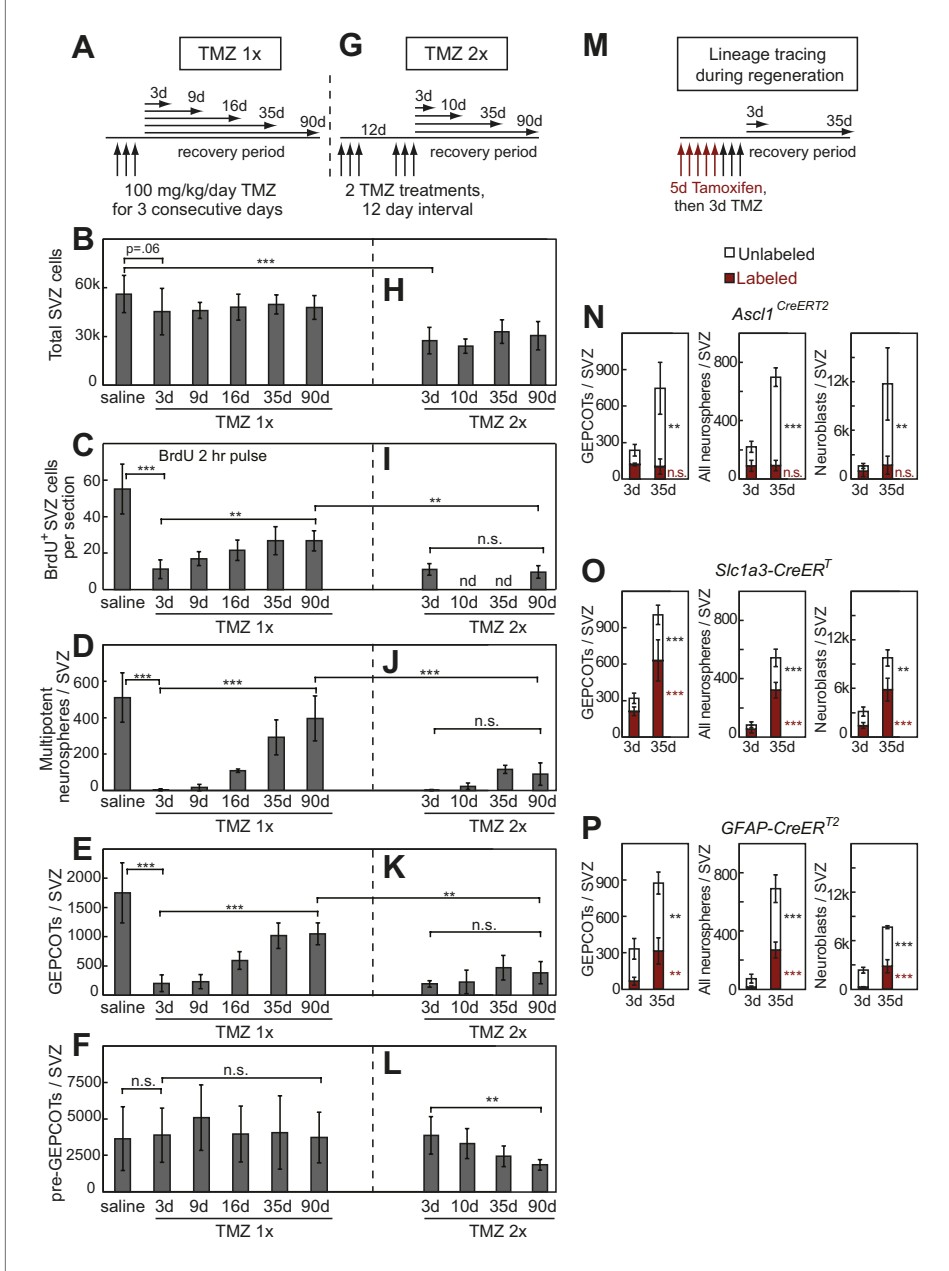

**Figure 5**. Treatment with temozolomide does not affect the frequency of pre-GEPCOT cells but ablates GEPCOT NICs. (**A–F**) Mice were injected with TMZ (100 mg/kg/day) for three consecutive days to ablate dividing cells, then allowed to recover for 3 to 90 days before analysis (TMZ 1x). All TMZ 1x data reflect 5–7 mice per time point from seven independent experiments. (**G–L**) Alternatively, mice were serially treated with two doses of TMZ 12 days apart then allowed to recover for 3 to 90 days before analysis (TMZ 2x). All TMZ 2x data reflect 4–9 mice per time point from six independent experiments. At each time point after TMZ treatment the panels show the total number of cells isolated per SVZ (**B** and **H**), the number of SVZ cells per section that incorporated a 2 hr pulse of BrdU (**C** and **I**), the number of multipotent neurospheres that arose in culture per SVZ (**D** and **J**), the number of GEPCOTs per SVZ (**E** and **K**), and the number of pre-GEPCOTs per SVZ (**F** and **L**). (**M–P**) Mice were injected with tamoxifen (80 mg/kg/day i.p.) for 5 days, then with TMZ (100 mg/kg/day i.p.) for 3 days, then recovered for 3 or 35 days to observe regeneration. The numbers of labeled GEPCOTs were measured after recombination with *Ascl1CreERT2* (**N**, 4 mice per condition from two independent experiments), *Slc1a3-CreERT* (**O**, 5–6 mice per condition from three independent experiments), or *GFAP-CreERT2* (**P**, 4 mice per condition from two independent experiments). All data represent mean ± SD. Statistical significance was tested with a one-way ANOVA followed by Sidak's post-hoc test

*Figure 5. Continued on next page*

*Figure 5. Continued*

for the indicated comparisons in **B**–**F** and **H**–**L**. Statistical significance in **N**–**P** was assessed with two-tailed student's *t* tests. *p<0.05, **p<0.01, ***p<0.001, n.s. not significant.

Consistent with the observation that pre-GEPCOT cells were sensitive to serial TMZ treatment, we detected little recovery of dividing SVZ cells (*Figure 5I*), NICs (*Figure 5J*), or GEPCOTs (*Figure 5K*) after serial TMZ treatment. These data suggest that GEPCOT NICs are replenished after ablation by pre-GEPCOT qNSCs.

To directly test whether GEPCOTs arise from pre-GEPCOT cells during SVZ regeneration we performed lineage tracing using multiple Cre alleles by first labeling cells with tamoxifen and then treating with TMZ (*Figure 5M*). After recombination of a conditional reporter with the GEPCOT marker *Ascl1*$^{CreERT2}$, the number of unlabeled GEPCOTs per SVZ increased from 120 ± 50 at 3 days after TMZ to 640 ± 210 at 35 days after TMZ (p<0.01) but *Ascl1*$^{CreERT2}$-labeled GEPCOTs did not significantly change (*Figure 5N*). The GEPCOTs, NICs, and neuroblasts that regenerated after TMZ treatment thus did not arise from *Ascl1*$^{CreERT2}$-labeled GEPCOTs (*Figure 5N*). In contrast, after recombination of a conditional reporter with the pre-GEPCOT markers *Slc1a3-CreER*$^T$ and *GFAP-CreER*$^{T2}$ we observed significant (p<0.05) increases in the frequencies of labeled GEPCOT cells, NICs, and neuroblasts between 3 and 35 days after TMZ treatment (*Figure 5O,P*). These results strongly suggest that pre-GEPCOT cells give rise to GEPCOTs, NICs, and neuroblasts during SVZ regeneration.

## Bmi-1 is required by pre-GEPCOT qNSCs and GEPCOT NICs in vivo

pre-GEPCOT qNSCs and GEPCOT NICs expressed similar levels of *Bmi-1* by qRT-PCR (*Figure 6A*). To study the molecular mechanisms that regulate the maintenance of pre-GEPCOT qNSCs and GEPCOT NICs in vivo we generated a floxed allele of *Bmi-1* (*Figure 6—figure supplement 1A,B*) and conditionally deleted *Bmi-1* using *Nestin-Cre* (*Tronche et al., 1999*). This allele of *Nestin-Cre* deletes broadly throughout the neuroepithelium by E10.5 such that there is nearly homogeneous recombination in the postnatal CNS (http://cre.jax.org/Nes/Nes-CreNano.html). Consistent with this, *Nestin-Cre; Bmi-1*$^{fl/fl}$ mice exhibited a loss of Bmi-1 protein in the cortex and SVZ by western blot (*Figure 6B*) and a loss of *Bmi-1* transcripts in SVZ cells, GEPCOT cells, and pre-GEPCOT cells by qRT-PCR (*Figure 6—figure supplement 1G*).

*Nestin-Cre; Bmi-1*$^{fl/fl}$ mice were modestly but significantly smaller than littermate controls (*Figure 6—figure supplement 1C*). Unlike germline *Bmi-1* deficient mice, which generally die within a month after birth (*van der Lugt et al., 1994*; *Lessard and Sauvageau, 2003*; *Park et al., 2003*; *Bruggeman et al., 2005*; *Molofsky et al., 2005*), we were able to age *Nestin-Cre; Bmi-1*$^{fl/fl}$ mice for up to 2 years. These mice did exhibit neurological deficits that worsened during aging, such as ataxia, as was reported in germline *Bmi-1* deficient mice (*van der Lugt et al., 1994*). However, the survival of *Nestin-Cre; Bmi-1*$^{fl/fl}$ mice throughout adulthood made it possible for the first time to test whether Bmi-1 is autonomously required by qNSCs or NICs in the adult brain.

Consistent with the phenotype of germline *Bmi-1* deficient mice (*van der Lugt et al., 1994*; *Bruggeman et al., 2005*; *Molofsky et al., 2005*), brain morphology appeared relatively normal in *Nestin-Cre; Bmi-1*$^{fl/fl}$ mice (*Figure 6C*, *Figure 6—figure supplement 1E*). Brain size was slightly but significantly reduced in *Nestin-Cre; Bmi-1*$^{fl/fl}$ mice as compared to littermate controls (*Figure 6—figure supplement 1D*) but was normal as a proportion of body mass (data not shown). The cerebellums from *Nestin-Cre; Bmi-1*$^{fl/fl}$ mice were smaller than in control mice and had significantly thinner molecular layers (*Figure 6D,E*). We also observed unusually prominent GFAP staining throughout the cortex of adult *Nestin-Cre; Bmi-1*$^{fl/fl}$ mice (*Figure 6—figure supplement 1F*). Otherwise, the olfactory bulb, hippocampus, and cortex appeared grossly normal (*Figure 6—figure supplement 1E*), though additional work will be required to carefully assess laminar organization and identity in each brain region. Our data suggest that the smaller cerebellums and prominent GFAP staining reflect a cell-autonomous requirement for Bmi-1 in fetal neural stem/progenitor cells.

We observed complete *Bmi-1* recombination in sorted SVZ cells, GEPCOT cells, and pre-GEPCOT cells isolated from adult *Nestin-Cre; Bmi-1*$^{fl/fl}$ mice (*Figure 6F*). 12 month-old *Nestin-Cre; Bmi-1*$^{fl/fl}$ mice exhibited a significantly reduced number of SVZ cells that incorporated a 2-hr pulse of BrdU (*Figure 6G*). We measured the rate of neurogenesis in *Nestin-Cre; Bmi-1*$^{fl/fl}$ mice and *Bmi-1*$^{fl/fl}$ controls by administering

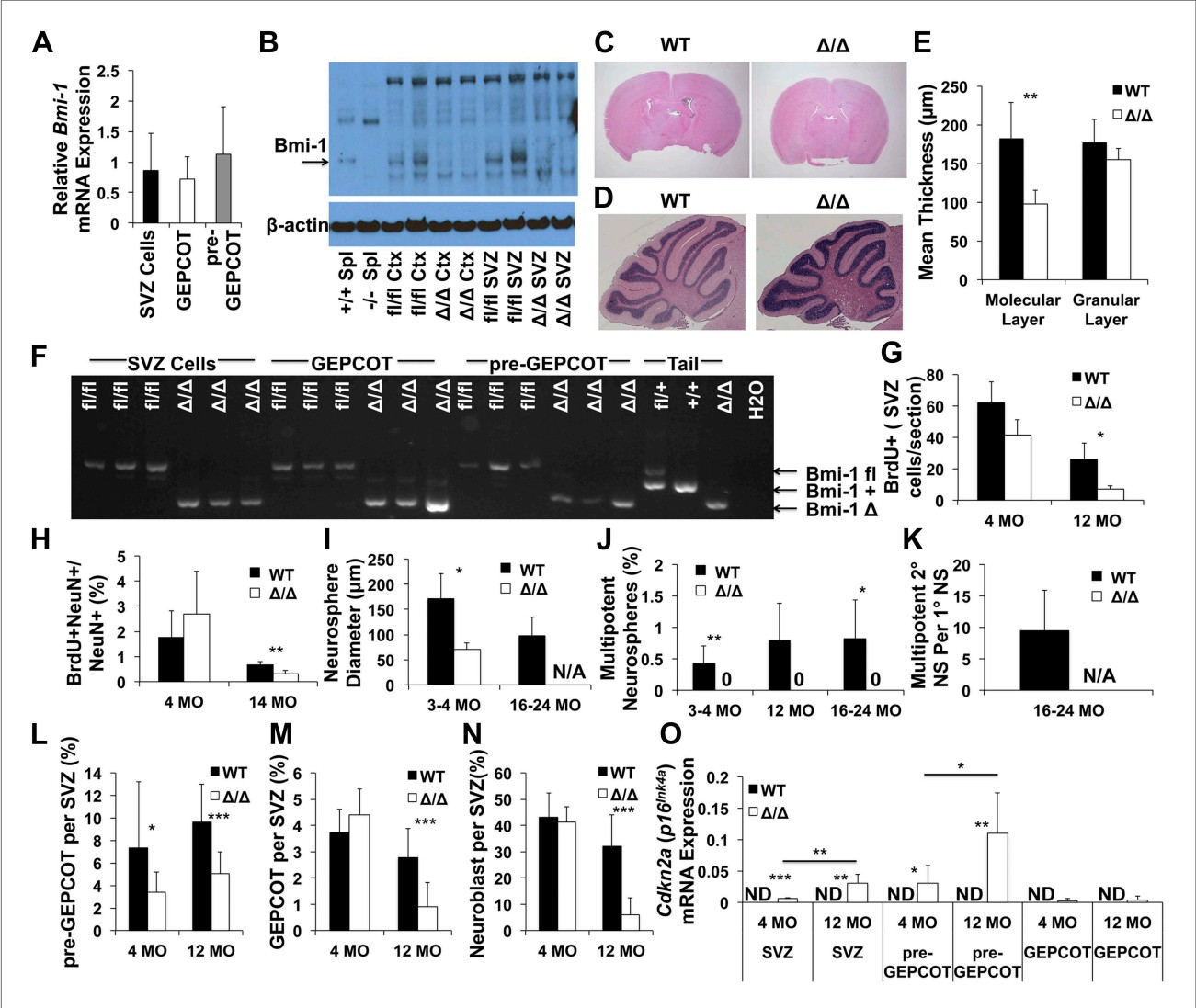

**Figure 6**. Reduced neurogenesis, SVZ cell proliferation, GEPCOT frequency, and pre-GEPCOT frequency in *Nestin-Cre; Bmi-1^fl/fl* adult mice. (**A**) *Bmi-1* transcript levels in GEPCOT and pre-GEPCOT cells, normalized to β-actin and shown relative to SVZ cells (n = 5 mice in two independent experiments). (**B**) Western blot of SVZ cells and cortical (Ctx) cells from adult *Nestin-Cre; Bmi-1^fl/fl* (Δ/Δ, n = 2) mice and *Bmi-1^fl/fl* (fl/fl, n = 2) controls. One of two independent blots is shown. Splenocytes (Spl) from wild-type (+/+) and germline *Bmi-1* deficient (−/−) mice are shown as controls. (**C–D**) Representative images of the forebrain (**C**) and cerebellum (**D**). (**E**) Cerebellar molecular and granular layer thickness (n = 3–6 mice/genotype with >8 measurements from two sections/mouse). (**F**) PCR analysis of genomic DNA from SVZ cells, GEPCOT cells, and pre-GEPCOT cells (3 mice/genotype, one of two independent experiments) as compared to controls (tail DNA and water). A faint non-specific band is visible in Δ/Δ GEPCOT cells. (**G**) The number of BrdU⁺ SVZ cells per section after a 2-hr pulse of BrdU (n = 3 mice/genotype/age, >6 sections per mouse). (**H**) The frequency of newborn olfactory bulb BrdU⁺NeuN⁺ neurons.after BrdU administration for 7 days followed by 4 weeks without BrdU (n = 3–6 mice/genotype/age in two independent experiments). (**I**) The diameter of primary neurospheres (n = 3–4 mice/genotype/age in four independent experiments. N/A: no neurospheres were formed). (**J**) The frequency of SVZ cells that formed multipotent neurospheres (n = 4–7 mice/genotype/age in four independent experiments). (**K**) The number of secondary neurospheres generated upon subcloning of individual neurospheres (n = 6 neurospheres/mouse for 3 WT mice). (**L–N**) The frequency of (**L**) pre-GEPCOT cells, (**M**) GEPCOT cells and (**N**) neuroblasts in the SVZ of 4- and 12-month-old mice (n = 13 mice/genotype/age in three independent experiments). (**O**) *Cdkn2a* (*p16^Ink4a^*) mRNA expression normalized to β-actin in SVZ, pre-GEPCOT and GEPCOT cells (n = 5 mice/genotype/age). *Cdkn2a* (*p16^Ink4a^*) was undetectable in all *Bmi-1^fl/fl* samples. In *Nestin-Cre; Bmi-1^fl/fl* mutants *Cdkn2a* (*p16^Ink4a^*) was detected in SVZ cells (5/5 4-month-old; 5/5 12-month-old) and pre-GEPCOT cells (4/5 4-month-old; 5/5 12-month-old) but usually not in GEPCOT cells (2/5 4-month-old; 1/5 12-month-old). Data represent mean ± SD. In **C–E** and **G–O**, WT is *Bmi-1^fl/fl* and Δ/Δ is *Nestin-Cre; Bmi-1^fl/fl* adult mice, and times are ages of mice. Statistical significance was assessed by two-tailed student's *t* tests. *p<0.05, **p<0.01, ***p<0.001.

The following figure supplements are available for figure 6:

**Figure supplement 1**. Targeting strategy for engineering a floxed allele of *Bmi-1* for conditional deletion.

BrdU for 7 days (beginning at 4 or 14 months of age) followed by a 4-week chase without BrdU. We quantified the frequency of BrdU$^+$NeuN$^+$ newborn neurons in sections from the olfactory bulb by microscopy. *Bmi-1* deletion had no effect on neurogenesis at 4 months of age, but significantly reduced olfactory bulb neurogenesis in 14-month-old mice (*Figure 6H*, p<0.01).

Neurospheres formed by *Nestin-Cre; Bmi-1$^{fl/fl}$* SVZ cells were significantly smaller than control neurospheres (*Figure 6I*), were unable to undergo multilineage differentiation (*Figure 6J*), and could not be passaged (*Figure 6K*). In contrast, control neurospheres readily underwent multilineage differentiation (*Figure 6J*) and were able to generate multipotent daughter neurospheres upon subcloning into secondary cultures (*Figure 6K*).

To assess whether *Bmi-1* deficiency affected pre-GEPCOT qNSCs or GEPCOT NICs in vivo we assessed the frequencies of these cells in SVZs from 4 and 12 month old *Nestin-Cre; Bmi-1$^{fl/fl}$* mice and littermate controls. The frequency of pre-GEPCOT cells was significantly reduced at 4 months of age (*Figure 6L*). We observed no significant effects on the frequencies of GEPCOT cells (*Figure 6M*) or neuroblasts (*Figure 6N*) at 4 months of age. However, each of these populations was significantly depleted in 12 month old *Nestin-Cre; Bmi-1$^{fl/fl}$* mice (*Figure 6L–N*). Bmi-1 is thus required for the maintenance of normal numbers of pre-GEPCOT qNSCs and GEPCOT NICs in the adult brain. Nonetheless, some pre-GEPCOT cells, GEPCOT cells, and neurogenesis did persist for at least a year in adult mice, demonstrating that NSCs are not absolutely dependent upon Bmi-1 for their maintenance in the adult brain.

Bmi-1 promotes the maintenance of neural stem cells partly by repressing the *Cdkn2a* (*p16$^{Ink4a}$*/*p19$^{Arf}$*) locus (*Jacobs et al., 1999*; *Molofsky et al., 2003*, *2005*; *Bruggeman et al., 2007*). We were unable to detect *Cdkn2a* (*p16$^{Ink4a}$*) transcripts in wild-type SVZ cells, pre-GEPCOT qNSCs, or GEPCOT NICs in 4-month-old (0/5) or 12 month old (0/5) mice (*Figure 6O*). In contrast, 4-month old *Bmi-1*-deficient SVZ cells (5/5 mice) and pre-GEPCOT cells (4/5 mice) did express *Cdkn2a* (*p16$^{Ink4a}$*) and the level of *Cdkn2a* (*p16$^{Ink4a}$*) expression significantly increased between 4 and 12 months of age (*Figure 6O*; detected in 5/5 mice for both populations). Low level *Cdkn2a* (*p16$^{Ink4a}$*) expression was detected in GEPCOT cells from two of five 4 month-old *Bmi-1*-deficient mice and in GEPCOT cells from one of five 12 month old *Bmi-1*-deficient mice (*Figure 6O*). Bmi-1 is thus required to repress *Cdkn2a* (*p16$^{Ink4a}$*) expression in pre-GEPCOT qNSCs.

### *Bmi-1* promotes adult neurogenesis and gliogenesis in vivo

To conditionally delete *Bmi-1* in the adult brain we used *Nestin-CreER$^{T2}$*, which deletes broadly throughout the SVZ, including within qNSCs that give rise to SVZ cells after AraC treatment (*Giachino and Taylor, 2009*). We observed significantly lower levels of *Nestin* transcripts within pre-GEPCOT qNSCs as compared to GEPCOT NICs (*Figure 7—figure supplement 1A*) and only 9.3 ± 7.3% of pre-GEPCOTs stained positively for Nestin protein as compared to 89.7 ± 8.5% of GEPCOTs and 39.0 ± 5.3% of SVZ cells (*Figure 7—figure supplement 1B*). We also observed higher levels of *Nestin* transgene expression in GEPCOT cells as compared to pre-GEPCOT cells, including *Nestin-mCherry* and *Nestin-GFP* (*Figure 7—figure supplement 1C,D*). However, when we analyzed recombination of a conditional reporter in *Nestin-CreER$^{T2}$* mice, we observed labeling of 99 ± 1.1% of GEPCOT cells and 96 ± 1.9% of pre-GEPCOT cells (*Figure 7—figure supplement 1E*). Therefore, although endogenous *Nestin* is expressed at lower levels in qNSCs as compared to NICs, *Nestin* second intronic enhancer transgenes are variably expressed in both pre-GEPCOT qNSCs and GEPCOT NICs, and *Nestin-CreER$^{T2}$* gives nearly complete recombination in both cell populations.

We administered tamoxifen in the chow of *Nestin-CreER$^{T2}$*; *Bmi-1$^{fl/fl}$* mice and littermate controls for 30 days beginning at 6 weeks of age and assessed recombination efficiency 2 weeks after completing tamoxifen treatment (i.e., starting at 3 months of age). Polymerase chain reaction (PCR) analysis of genomic DNA from individual neurospheres revealed complete deletion of *Bmi-1* in at least 95% of neurospheres (*Figure 7—figure supplement 2A,B*). Western blot analysis demonstrated a near total loss of Bmi-1 protein from neurospheres (*Figure 7A*), consistent with the high rate of recombination in individual neurospheres. By PCR analysis of genomic DNA we observed near complete recombination in unfractionated SVZ cells and complete recombination in pre-GEPCOT and GEPCOT cells isolated from adult *Nestin-CreER$^{T2}$*; *Bmi-1$^{fl/fl}$* mice (*Figure 7B*).

Body mass (*Figure 7—figure supplement 2C*), brain histology (data not shown), and cerebellum size (*Figure 7—figure supplement 2D*) were all grossly normal in *Nestin-CreER$^{T2}$*; *Bmi-1$^{fl/fl}$* mice six months after tamoxifen treatment. To assess the consequences of *Bmi-1* deletion from adult neural

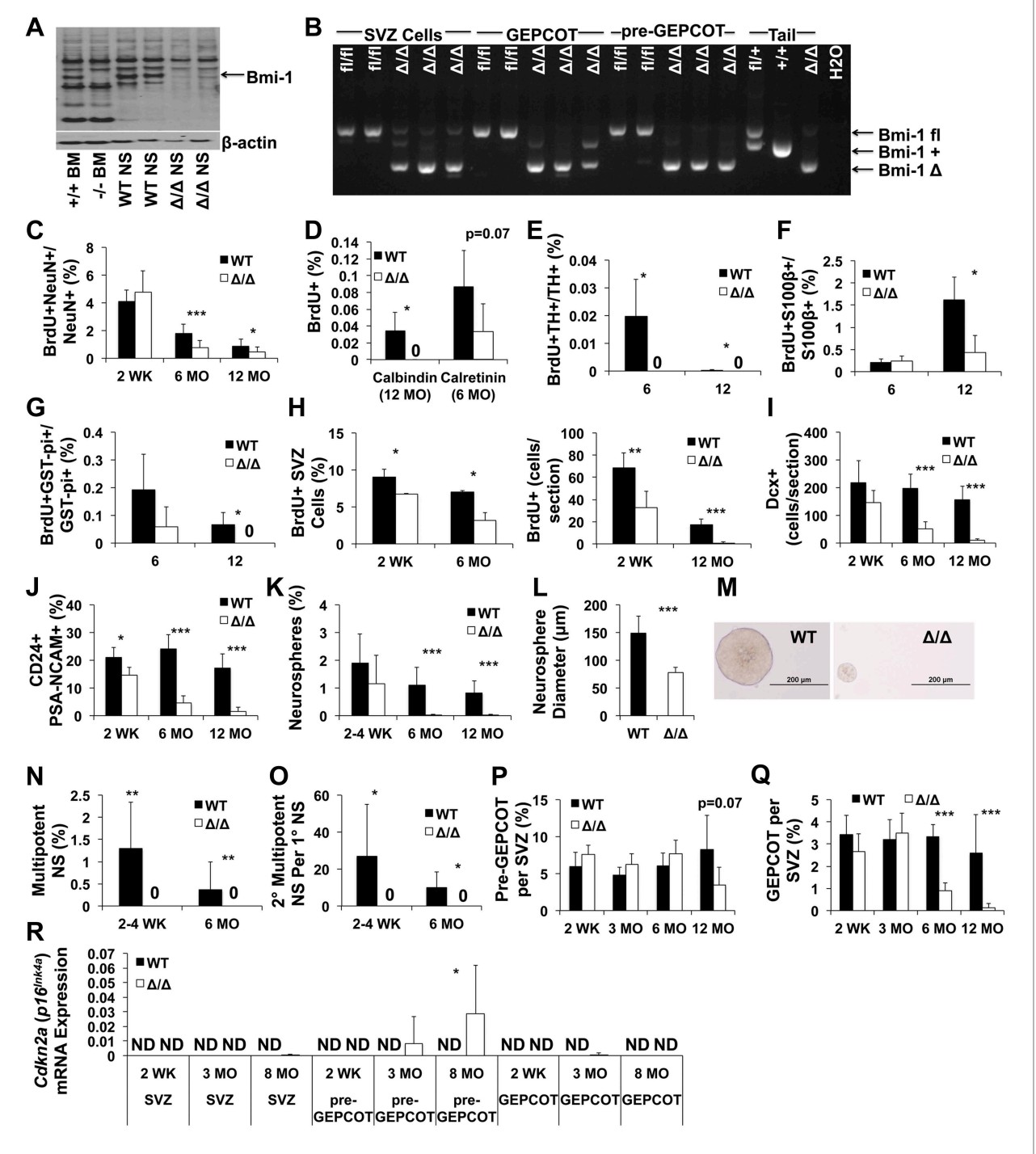

**Figure 7**. Reduced neurogenesis, gliogenesis, SVZ cell proliferation, and GEPCOT NICs in adult *Nestin-CreER[T2]; Bmi-1[fl/fl]* mice relative to littermate controls. (**A**) Western blot of pooled neurospheres (NS) cultured at 2 weeks after tamoxifen treatment (n = 2 mice/genotype). Wild-type and germline *Bmi-1[−/−]* bone marrow (BM) cells are shown as controls. (**B**) PCR analysis of genomic DNA from SVZ cells, GEPCOT cells, and pre-GEPCOT cells 1 day after tamoxifen treatment as compared to controls (tail DNA and water) (n = 2–3 mice/genotype). A faint non-specific product is visible in lanes 3, 8 and 10. (**C**) The frequency of newborn BrdU⁺NeuN⁺ neurons in the olfactory bulb (n = 5–11 mice/genotype/timepoint in 10 independent experiments). (**D–G**) The frequencies of newborn BrdU⁺ Calbindin⁺ (**D**), Calretinin⁺ (**D**), Tyrosine Hydroxylase⁺ (**E**), and S100β⁺ (**F**) cells in the olfactory bulb or GST-pi⁺ (**G**) cells in the cortex (n = 3–5 mice/genotype/timepoint in two independent experiments). For panels **C–G** BrdU was administered for 7 days followed by 4 weeks without BrdU. (**H**) The frequency (left panel) or number (right panel) of BrdU⁺ SVZ cells per section after a 2-hr pulse of BrdU (n = 3–11 mice/genotype/timepoint in six independent experiments). (**I–J**) The number of Dcx⁺ neuroblasts per section (**I**) or frequency of CD24^mid PSA-NCAM⁺
*Figure 7. Continued on next page*

*Figure 7. Continued*

neuroblasts as determined by flow cytometry (n = 4–11 mice/genotype/age in 6–7 independent experiments). (**K** and **N**) The frequencies of SVZ cells that formed neurospheres (>50 μm; **K**), or multipotent neurospheres (N; n = 5–9 mice/genotype/timepoint in six independent experiments). (**L**) The diameter of primary neurospheres from mice 2–4 weeks after tamoxifen treatment (n = 7–9 mice/genotype in two independent experiments). (**M**) Representative primary neurospheres from mice 2 weeks after tamoxifen treatment. (**O**) The number of multipotent secondary neurospheres generated upon subcloning of individual primary neurospheres (n = 3–9 mice/genotype/timepoint in three independent experiments). (**P** and **Q**) The frequency of pre-GEPCOT cells (**P**) or GEPCOT cells (**Q**) in the SVZ (n = 3–11 mice/genotype/timepoint in seven independent experiments). (**R**) qRT-PCR analysis of *Cdkn2a* (*p16^Ink4a*) transcript levels expression normalized to β-actin in SVZ, pre-GEPCOT and GEPCOT cells (n = 5–9 mice/genotype/timepoint). All data represent mean ± SD. In **A** and **C–R**, WT is *Bmi-1^fl/fl* and Δ/Δ is *Nestin-CreER^T2*; *Bmi-1^fl/fl* adult mice, and timepoints are times after tamoxifen induction beginning at 6 weeks of age. Statistical significance was assessed with two-tailed student's *t* tests. *p<0.05, **p<0.01, ***p<0.001.

The following figure supplements are available for figure 7:

**Figure supplement 1**. *Nestin* expression in GEPCOT and pre-GEPCOT cells.

**Figure supplement 2**. Adult *Nestin-CreER^T2*; *Bmi-1^fl/fl* mice show *Bmi-1* recombination in pre-GEPCOT qNSCs.

stem/progenitor cells, we measured the rate of neurogenesis in *Nestin-CreER^T2*; *Bmi-1^fl/fl* mice and *Bmi-1^fl/fl* controls by administering BrdU for 7 days (beginning at 2 weeks, 6 months and 12 months after tamoxifen treatment, at which time the mice were approximately 3 months old, 9 months old and 15 months old, respectively) followed by a 4-week chase without BrdU. *Bmi-1* deficiency had no effect on the frequency of BrdU+NeuN+ newborn neurons in the olfactory bulb 2 weeks after tamoxifen treatment but significantly reduced neurogenesis 6 and 12 months after tamoxifen (*Figure 7C*, p<0.05). The reduced neurogenesis in *Bmi-1* mutant mice was evident in all subsets of neurons that we investigated, including BrdU+Calretinin+ neurons, BrdU+Calbindin+ neurons, and BrdU+Tyrosine Hydroxylase+ neurons (*Figure 7D,E*). *Bmi-1* is therefore required in adult neural stem/progenitor cells for normal neurogenesis but *Bmi-1* deficiency did not completely eliminate the generation of forebrain neurons during the first year of life.

Consistent with prior studies, the vast majority of the newborn cells in the adult olfactory bulb were neurons, but the small numbers of astrocytes that arose in the olfactory bulb and oligodendrocytes that arose in the cortex were also diminished after *Bmi-1* deletion. The frequency of newborn olfactory bulb BrdU+S100β+ astrocytes appeared normal 6 months after tamoxifen treatment but was significantly reduced 12 months after tamoxifen in *Bmi-1* mutant mice (*Figure 7F*). The frequency of newborn cortical BrdU+GST-pi+ oligodendrocytes declined 6 months after tamoxifen treatment and could no longer be detected after 12 months (*Figure 7G*). The deficits in neurogenesis and gliogenesis became worse over time after *Bmi-1* deletion.

We quantified the frequency of SVZ cells that incorporated a 2-hr pulse of BrdU. *Bmi-1* mutant mice had significant reductions in the frequencies of BrdU+ SVZ cells at 2 weeks (50% reduction, p<0.02), 6 months (55% reduction, p<0.02) and 12 months (90% reduction, p<0.001) after tamoxifen treatment (*Figure 7H*). *Bmi-1* mutant mice also had significant (p<0.001) reductions in the frequencies of Mcm2+ (*Figure 7—figure supplement 2E,G*) and Ki67+ (*Figure 7—figure supplement 2F*) SVZ cells. Consistent with this, *Bmi-1* mutant mice exhibited significant (p<0.001) declines in the frequencies of Dcx+ (*Figure 7I*) and CD24midPSA-NCAM+ neuroblasts (*Figure 7J*) 6 and 12 months after tamoxifen treatment.

## GEPCOT NICs acutely depend upon Bmi-1 in the adult SVZ

To assess whether *Bmi-1* is required by adult NICs we plated SVZ cells from tamoxifen-treated *Nestin-CreER^T2*; *Bmi-1^fl/fl* and control mice in non-adherent cultures and assessed GEPCOT cell frequency. 2 weeks after tamoxifen treatment the frequencies of NICs (*Figure 7K*) and GEPCOTs (*Figure 7Q*) were unchanged in *Bmi-1* mutant mice as compared to littermate controls. However, the *Bmi-1*-deficient neurospheres were significantly smaller than control neurospheres (*Figure 7L,M*) and did not undergo multilineage differentiation (*Figure 7N*; they formed only astrocytes upon transfer to adherent cultures) or self-renew upon subcloning (*Figure 7O*). At 6 and 12 months after tamoxifen treatment, *Bmi-1* deficient SVZ cells formed virtually no neurospheres (*Figure 7K*). Consistent with this, *Nestin-CreER^T2*; *Bmi-1^fl/fl* mice also had significantly lower frequencies of GEPCOTs than control mice at 6 months and 12 months after tamoxifen treatment, and the magnitude of the depletion increased over time

(*Figure 7Q*). Virtually no GEPCOTs could be found in *Nestin-CreER^T2*; *Bmi-1^fl/fl* mice 12 months after tamoxifen treatment (*Figure 7Q*). These data demonstrate that GEPCOT NICs require *Bmi-1* in vivo for their maintenance in the adult SVZ, though the cells are able to persist for several months after *Bmi-1* deletion before they are completely eliminated.

To test whether qNSCs depend upon Bmi-1 for their maintenance in vivo we examined the frequency of pre-GEPCOT cells in tamoxifen-treated *Nestin-CreER^T2*; *Bmi-1^fl/fl* mice and littermate controls. pre-GEPCOT cells were not depleted at 2 weeks or 6 months after tamoxifen treatment (*Figure 7P*). We observed a clear trend toward reduced pre-GEPCOT cell frequency in *Nestin-CreER^T2*; *Bmi-1^fl/fl* mice 12 months after tamoxifen treatment but the effect was not statistically significant.

To test whether there was a loss of Bmi-1 function in these cells we performed qRT-PCR to assess *Cdkn2a* (*p16^Ink4a*) expression in cells from *Nestin-CreER^T2*; *Bmi-1^fl/fl* mice. One mechanism by which Bmi-1 promotes the maintenance of neural stem/progenitor cells is by negatively regulating *Cdkn2a* (*p16^Ink4a*) expression (*Jacobs et al., 1999*; *Molofsky et al., 2003*; *Bruggeman et al., 2005*; *Molofsky et al., 2005*). We did not detect *Cdkn2a* (*p16^Ink4a*) transcripts in wild-type SVZ cells, pre-GEPCOT qNSCs, or GEPCOT NICs 2 weeks (0/5), 3 months (0/6), or 8 months (0/7) after tamoxifen treatment (*Figure 7R*). We rarely detected *Cdkn2a* (*p16^Ink4a*) transcripts in SVZ cells or GEPCOT cells from *Nestin-CreER^T2*; *Bmi-1^fl/fl* mice 2 weeks (0/5 SVZ, 0/5 GEPCOT), 3 months (0/5 SVZ, 1/5 GEPCOT) or 8 months (1/9 SVZ, 0/9 GEPCOT) after tamoxifen treatment. *Cdkn2a* (*p16^Ink4a*) transcripts were rarely detected in pre-GEPCOT cells from *Nestin-CreER^T2*; *Bmi-1^fl/fl* mice 2 weeks (0/5) or 3 months (1/5) after tamoxifen treatment. However, 8 months after tamoxifen treatment *Cdkn2a* (*p16^Ink4a*) expression was detected in pre-GEPCOT cells from seven of nine *Nestin-CreER^T2*; *Bmi-1^fl/fl* mice (*Figure 7R*, *Figure 7—figure supplement 2H*). These data demonstrate that Bmi-1 is required during adulthood to negatively regulate *Cdkn2a* (*p16^Ink4a*) expression in pre-GEPCOT qNSCs but that these cells are depleted more slowly than GEPCOT NICs in the adult SVZ. Although high levels of *Cdkn2a* (*p16^Ink4a*) expression are associated with cellular senescence, we were unable to detect any increase in senescence associated β-galactosidase activity in pre-GEPCOT cells from 14 month old *Nestin-Cre; Bmi-1^fl/fl* mice or in cultured neural stem/progenitor cells from *Bmi-1* germline deficient mice (data not shown).

## Discussion

By screening almost 400 antibodies against distinct cell surface antigens we identified two phenotypically and functionally distinct populations of neural stem/progenitor cells from the adult mouse SVZ (*Figure 8*). GEPCOT cells were highly enriched for NICs (*Figure 1C*) and highly mitotically active in vivo (*Figure 1D*) but persisted only transiently in the SVZ based on fate mapping with *Ascl1^CreERT2* or *Dlx1^CreERT2* (*Figure 2B,C*). In contrast, pre-GEPCOT cells lacked the ability to form neurospheres or adherent colonies in culture (*Figure 4B*), and were quiescent in vivo (*Figure 4C*) but were long-lived in the SVZ based on fate mapping with the stem cell markers *Slc1a3-CreER^T* (*Wang et al., 2012*), *GFAP-CreER^T2* (*Giachino et al., 2013*), *Sox2^CreERT2* (*Arnold et al., 2011*), and *Gli1^CreERT2* (*Ahn and Joyner, 2005*; *Lee et al., 2012*) (*Figure 4F–J*). In contrast to GEPCOT NICs, pre-GEPCOT cells were resistant to TMZ (*Figure 5E,F*). Although TMZ eliminated virtually all NICs from the SVZ (*Figure 5D*), the dividing cells, NICs, and GEPCOTs regenerated within a month of TMZ treatment (*Figure 5*). The persistence of pre-GEPCOT cells after TMZ (*Figure 5F*), and the regeneration of GEPCOTs from a precursor that expresses *Slc1a3-CreER^T* and *GFAP-CreER^T2* but not *Ascl1^CreERT2* (*Figure 5N–P*) suggest that the SVZ regenerates from pre-GEPCOT qNSCs.

The conclusion that pre-GEPCOT cells include qNSCs is also supported by the sustained contribution of *GFAP-CreER^T2* marked cells to the SVZ, as *GFAP-CreER^T2* labeled many pre-GEPCOT cells but few GEPCOT cells or NICs immediately after tamoxifen treatment (*Figure 3A*, *Figure 4I*). Thus, our data demonstrate that *GFAP-CreER^T2* recombination is a marker of qNSCs that survive treatment with anti-mitotic agents and contribute to an increasing proportion of pre-GEPCOT cells, GEPCOT cells, and neuroblasts over time (*Figures 4I and 5M–P*).

NSCs are commonly estimated to account for a few percent of cells in germinal zones in the adult mouse brain (*Doetsch et al., 1997*; *Mirzadeh et al., 2008*; *Pastrana et al., 2009*). Although pre-GEPCOT cells accounted for 6 ± 3% of SVZ cells, this population is likely to be heterogeneous, consistent with its heterogeneous GFAP expression (*Figure 4A*). Thus, the actual frequency of qNSCs in the SVZ may be lower than 6%. The current lack of a clonal assay for qNSCs makes it impossible to test the purity of this population. In an effort to identify culture conditions permissive for colony formation by pre-GEPCOT cells we screened 36 different growth factors or medium supplements but none significantly increased

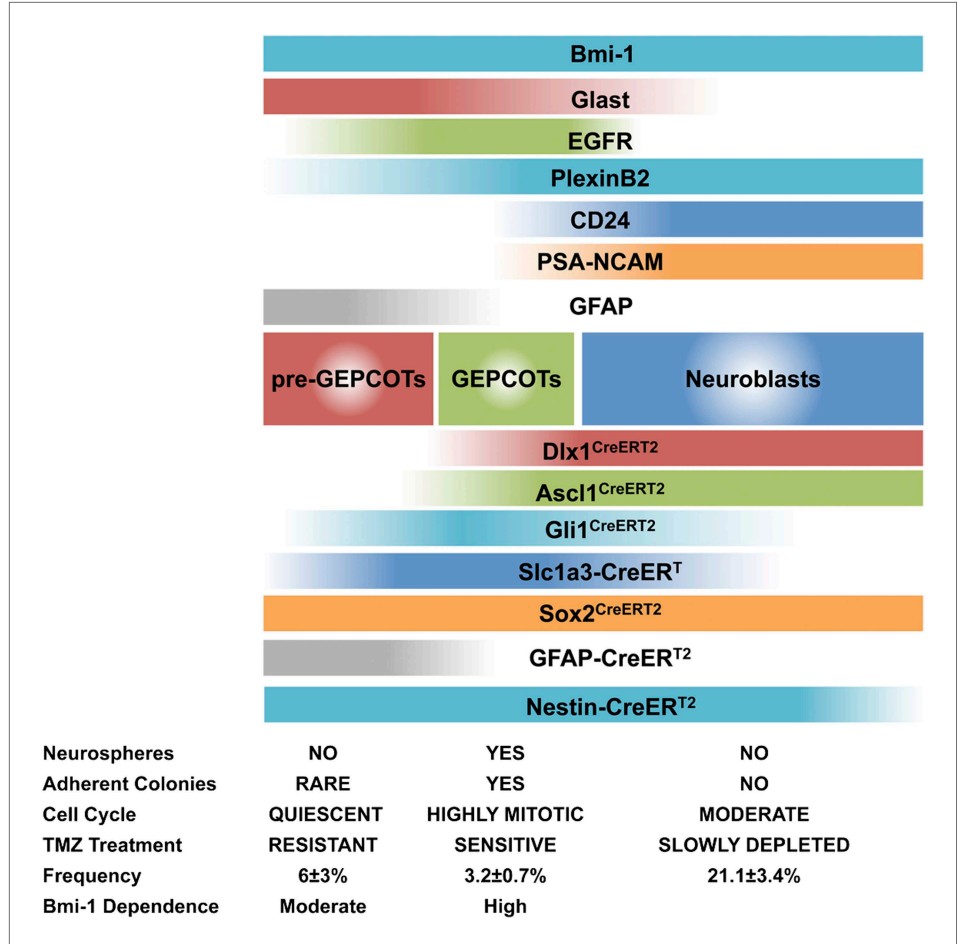

**Figure 8**. Phenotypically and functionally distinct populations of pre-GEPCOT qNSCs and GEPCOT NICs. Glast^high EGFR^−/low PlexinB2^mid CD24^−/low O4/PSA-NCAM^−/low Ter119/CD45^− pre-GEPCOT cells are a quiescent, TMZ resistant population containing qNSCs that give rise to Glast^mid EGFR^high PlexinB2^high CD24^−/low O4/PSA-NCAM^−/low Ter119/CD45^− GEPCOT cells that are highly mitotically active in vivo and enriched for NICs. pre-GEPCOT cells include Type B1 cells, GEPCOT cells include Type C cells, and neuroblasts make up Type A SVZ cells (*Doetsch et al., 1997*; *Doetsch et al., 1999a*). Bars for different *Cre* alleles represent the extent of recombination observed after 5 consecutive days of tamoxifen treatment followed by a 2 day chase.

colony formation by SVZ cells from normal or TMZ-treated mice (*Supplementary file 1B*). Thus, future studies will be required to develop culture conditions that permit efficient colony formation by individual qNSCs and to isolate these cells at high purity.

An interesting question for future studies will be whether there are regional differences in the distributions or properties of pre-GEPCOT qNSCs or GEPCOT NICs within the SVZ, corresponding to regional differences in stem/progenitor cell properties (*Merkle et al., 2007, 2014*; *Ihrie et al., 2011*).

Germline deficiency for the polycomb family member *Bmi-1* eliminates multipotent NICs and reduces proliferation and neurogenesis in the SVZ (*Molofsky et al., 2003*; *Bruggeman et al., 2005*; *Molofsky et al., 2005*; *Zencak et al., 2005*; *Bruggeman et al., 2007*; *Fasano et al., 2009*). However, the lack of a floxed allele of *Bmi-1*, the death of germline *Bmi-1*-deficient mice before adulthood, and the lack of prospective markers for NSCs made it impossible to directly test whether Bmi-1 was required for the maintenance of adult NSCs in vivo.

Conditional deletion of *Bmi-1* from the fetal and adult SVZ using *Nestin-Cre* (*Figure 6*) and *Nestin-CreR^T2* (*Figure 7*) demonstrated that pre-GEPCOT qNSCs and GEPCOT NICs require Bmi-1 to be maintained in normal numbers in the adult forebrain, but the depletion of these cells and the loss of neurogenesis occurred much more gradually than expected. Even when *Bmi-1* was completely deleted

in neural stem/progenitor cells during fetal development, normal or near-normal numbers of pre-GEPCOT cells, GEPCOT cells, and SVZ neuroblasts were found in the SVZ of 4 month old *Nestin-Cre*; *Bmi-1*[fl/fl] mice (*Figure 6L–N*). This demonstrates that pre-GEPCOT qNSCs and GEPCOT NICs can persist throughout fetal and postnatal development, and into adulthood, in the absence of *Bmi-1*. This suggests that qNSCs and NICs are less acutely dependent upon *Bmi-1* for their maintenance than expected based on studies of *Bmi-1* germline knockout mice (*van der Lugt et al., 1994*; *Molofsky et al., 2003*; *Bruggeman et al., 2005*; *Molofsky et al., 2005*; *Fasano et al., 2009*).

There are likely several reasons why *Bmi-1* germline knockout mice appeared to exhibit a more severe loss of neural stem/progenitor cells and neurogenesis than is evident in conditional knockout mice. First, the phenotype in *Bmi-1* germline knockout mice may indeed be more severe as non-cell-autonomous effects of *Bmi-1* deficient stroma contribute to stem cell deficits in these mice (*Oguro et al., 2006*, *2010*). Second, the death of *Bmi-1* germline knockout mice before young adulthood meant that adult phenotypes could not be studied directly and it was assumed that the deficits observed in the early postnatal period would rapidly worsen over time. Third, without the ability to prospectively identify qNSCs or NICs in vivo, these cells could only be studied based on neurosphere formation in culture. Since p16$^{Ink4a}$ and p19$^{Arf}$ expression are induced to a greater extent in culture than in vivo, the severity of *Bmi-1* deficiency phenotypes in the nervous system are sometimes exaggerated in culture (*Molofsky et al., 2005*; *He et al., 2009*). Our data indicate that *Bmi-1* is not required for the persistence of NSCs or neurogenesis into adulthood, though both are depleted over time during adulthood after *Bmi-1* deletion.

Consistent with the gradual depletion of pre-GEPCOT cells, GEPCOT cells, and neurogenesis during adulthood after conditional Bmi-1 deletion, we also observed a gradual increase in *Cdkn2a* (*p16$^{Ink4a}$*) expression in pre-GEPCOT cells but not in GEPCOT cells (*Figures 6O and 7R*, *Figure 7—figure supplement 2H*). This suggests that qNSCs are particularly dependent upon Bmi-1 for the repression of *Cdkn2a* (*p16$^{Ink4a}$*) expression.

The identification of markers that prospectively identify and isolate distinct populations of qNSCs and NICs will make it possible to characterize their properties in vivo rather than relying upon colony-forming assays in culture. Use of commercially available antibodies against cell surface antigens will enable such studies in a wide range of genetic backgrounds. Our data on Bmi-1 demonstrate how the existence of these markers makes it possible to assess the function of gene products within the neural stem/progenitor cell pool with a more granular appreciation for effects on qNSCs vs NICs.

## Materials and methods

### Mice

C57Bl/6 were maintained in standard cages with water and standard diet (Teklad 2916) *ad libitum*. *Rosa26$^{CAG-loxp-Stop-loxp-tdTomato(Ai14)}$* (referred to here as *Rosa26-loxp-tdTomato*) (*Madisen et al., 2010*), *Gli1$^{CreERT2}$* (*Ahn and Joyner, 2005*), *Sox2$^{CreERT2}$* (*Arnold et al., 2011*), *Dlx1$^{CreERT2}$* (*Taniguchi et al., 2011*), *Tg(Slc1a3-CreER$^T$)* (*Wang et al., 2012*), *Tg(Nestin-Cre)* (*Tronche et al., 1999*) and *Tg(Ubc-GFP)* (*Schaefer et al., 2001*) mice were obtained from The Jackson Laboratory (Bar Harbor, ME). *Tg(GFAP-CreER$^{T2}$)* (*Hirrlinger et al., 2006*) mice were provided by Frank Kirchhoff. *Tg(Nestin-GFP)* (*Birbrair et al., 2011*) and *Tg(Nestin-mCherry)* (*Ding et al., 2012*) mice were kindly provided by Grigori Enikopalov. *Tg(Nestin-CreER$^{T2}$)* mice were provided by G Fishell (*Balordi and Fishell, 2007*). *Ascl1$^{CreERT2}$* (*Kim et al., 2011*) mice were housed in the Johnson laboratory. All mice were backcrossed onto a C57BL/Ka background for at least three generations prior to analysis. For BrdU pulses up to 24 hr, 100 mg of BrdU/kg body mass dissolved in PBS was injected i.p. every 6 hours. For 2-week pulses of BrdU, mice were initially injected with 100 mg of BrdU/kg body mass then maintained on drinking water that contained 1 mg/ml BrdU until sacrifice.

Targeting vectors to generate *Bmi1$^{fl}$* mice (*Figure 6—figure supplement 1A*) were constructed by recombineering (*Liu et al., 2003*). W4 ES cells were electroporated with the targeting vector and positive clones identified by southern blotting were injected into blastocysts from C57BL/6-Tyr$^{c-2J}$ mice. The resulting male chimeric mice were bred to female C57BL/6-Tyr$^{c-2J}$ mice to obtain germline transmission. The FRT-Neo-FRT cassette was removed by crossing with Flp deleter mice (*Rodriguez et al., 2000*), and then *Bmi1$^{fl}$* mice were backcrossed onto a C57BL/Ka background for at least 10 generations. Animal protocols were approved by the University of Michigan Committee on the Use

and Care of Animals and the UT Southwestern Medical Center Institutional Animal Care and Use Committee (protocol# 2011-0104).

## Induction of recombination using tamoxifen

Tamoxifen (T5648; Sigma, St. Louis, MO) was dissolved in 90% corn oil/10% ethanol at 20 mg/ml, and injected at 80 mg/kg/day i.p. for 5 consecutive days into 8–12 week-old mice, then the mice were chased for 2, 7, 28, or 60 days until analysis. For *Bmi-1* experiments, tamoxifen citrate (Sigma or Spectrum Chemical, New Brunswick, NJ) was given in chow at 400 mg/kg with 5% sucrose (Harlan) to 6 week-old mice for 30 days. Mice were then fed a standard diet for at least 14 days before analysis.

## TMZ regeneration assay

TMZ (T2577; Sigma) was dissolved in 25% DMSO/75% 0.9% saline solution at 10 mg/ml by heating briefly to 90–100°C, then shaking and rapidly cooling. Mice were injected at 100 mg/kg/day for 3 consecutive days then allowed to recover for 3–90 days before analysis. Mice with enlarged spleen or thymus (sometimes present at 90 days after TMZ treatment) were excluded from analysis.

## SVZ cell preparation

SVZs from adult (8–60 week old) mice were dissected as described (*Mirzadeh et al., 2010*). SVZs were minced and digested with 300 µl of trypsin solution (Ca and Mg-free HBSS, 10 mM HEPES, 0.5 mM EDTA, 0.25 mg/ml trypsin (EMD Millipore, Billerica, MA), 10 µg/ml DNase I (Roche, Basel, Switzerland), pH 7.6) at 37°C for 20 min. Digestion was quenched with three volumes of staining medium (440 ml Leibovitz L-15 medium, 50 ml water, 5 ml 1M HEPES pH 7.3–7.4, 5 ml 100x Pen-Strep, 20 ml 77.7 mM EDTA pH 8.0 [prepared from $Na_2H_2EDTA$], 1 g bovine serum albumin [A7030; Sigma]) containing 100 µg/ml trypsin inhibitor (T6522; Sigma) and 10 µg/ml DNase I (Roche).

Digested SVZ pieces were centrifuged (220×*g*, 4 min, 4°C) then fresh staining medium was added and the pieces were triturated in 300 µl by gently drawing into a P1000 pipetman and expelling 25 times without forming bubbles. The cell suspension was then filtered through a 45-micron mesh, counted on a hemocytometer, and added to culture or processed for flow cytometry. Neurosphere formation, self-renewal, and differentiation assays were performed as described previously (*Molofsky et al., 2003*; *Nishino et al., 2008*), except that in some cases 10 µM Y-27632 hydrochloride (Rho-associated protein kinase inhibitor; Tocris Biosciences 1254, Bristol, UK) and 20 ng/ml IGF-1 (291-G1; R&D Systems, Minneapolis, MN) were also added to the culture medium because they were found to promote clonogenesis after sorting adult SVZ cells (data not shown). For adherent colony formation, *Tg(Ubc-GFP)* SVZ cells were co-cultured with ~80–90% confluent nontransgenic neonatal SVZ-derived astrocytes isolated and passaged twice prior to the experiment. GFP fluorescence was used to assess colony formation after 12–20 days in culture.

## Flow cytometry and sorting

For screening for markers of NICs, typically SVZs from 5–10 mice were pooled and stained with antibodies then analyzed using a 4 laser FACSAria III (Becton Dickenson, Franklin Lakes, NJ). Markers that stained SVZ cells heterogeneously were used to separate SVZ cells by flow cytometry into different fractions then sorted into non-adherent cultures to measure the frequency of NICs.

To stain SVZ cells with the combination of markers used to isolate GEPCOT or pre-GEPCOT cells, dissociated SVZ cells were centrifuged (220×*g*, 4 min, 4°C) and resuspended in 100 µl staining medium per brain. Then the following antibodies were added: BV421 anti-Ter119 (116233; 1/100; Biolegend, San Diego, CA), BV421 anti-CD45 (103133; 1/100; Biolegend), APC-eFluor 780 anti-CD24 (47-0242-82; 1/200; eBioscience, San Diego, CA), 5A5 ascites (anti-PSA-NCAM; Developmental Studies Hybridoma Bank, Iowa City, IA, 1/100), O4 ascites (1/200), anti-PlexinB2 (14-5665-82; 1/100; eBioscience), biotinylated anti-EGFR (BAF1280, 1/200; R&D Systems), anti-Glast (130-095-822; 1/10; Miltenyi Biotec, Bergisch Gladbach, Germany), and DAPI (50 µg/ml, 1/100). After adding antibodies, the cells were incubated on ice for 45 min, then washed with 2.5 ml fresh staining medium and pelleted (220g for 4 min at 4°C) and resuspended in secondary antibodies. Typically we used the following combination: APC anti-mouse IgG2a (115-135-206; 1/200; Jackson Immunoresearch, West Grove, PA), PE anti-hamster IgG (127-115-160; 1/200; Jackson Immunoresearch), PE-Cy7 anti-mouse IgM (25-5790-82; 1/100; eBioscience), BV605-Streptavidin (563260; 1/200; BD Biosciences). However when analyzing mice expressing tdTomato we used the following alternative secondary antibodies: PerCP-eFluor 710

anti-mouse IgG (46-4010-82; 1/100; eBioscience), APC anti-hamster IgG (127-135-160; 1/200; Jackson Immunoresearch), PE-Cy7 anti-mouse IgM (25-5790-82; 1/100; eBioscience), BV605-Streptavidin (563260; 1/200; BD Biosciences). Cells were stained with secondary antibodies on ice for 45 min, then washed and pelleted as above, and analyzed on the FACSAria in a volume of 200 µl per brain.

When sorting cells onto slides, a drop of 8 µl staining medium was placed on a SuperFrost Plus slide, and 100–300 cells were sorted directly into the drop. The slide was then kept humid for 45–60 min at ambient temperature to allow the cells to attach to the glass, then the cells were fixed with 4% PFA (20 min, RT). Cells were then permeabilized and stained with rat anti-BrdU (Abcam, Cambridge, MA, clone Bu1/75, 1/500, after heat-mediated antigen retrieval), mouse anti-Nestin (BD clone Rat-401, 1/100 after heat-mediated antigen retrieval), and rabbit anti-GFAP (Dako, Carpinteria, CA, 1/3000) antibodies using standard techniques.

## Immunostaining

Brains were fixed overnight at 4°C in 4% PFA in PBS, then cryoprotected in 30% sucrose in PBS for 1–3 days at 4°C, then frozen in OCT or cryogel on dry ice after 3–12 hr equilibration at 4°C. Sections were cut at 12 µm thickness spanning the rostral half of the SVZ (typically 6 sections per slide) or the entire olfactory bulb (typically 8 sections per slide). Sections were immunostained with the following primary antibodies: rat anti-BrdU (Abcam clone Bu1/75, 1/500, after heat-mediated antigen retrieval), guinea pig anti-Dcx (1/1000; Millipore), mouse anti-Mcm2 (BD Biosciences, 1/500, after heat-mediated antigen retrieval), rat anti-Ki67 (1/500; eBioscience), mouse anti-tyrosine hydroxylase (1/1000; Millipore), rabbit anti-calretinin (1/1000; Sigma), rabbit anti-calbindin (1/500; Millipore), rabbit anti-S100β (1/1000; Dako), rabbit anti-GST-pi (1/3000; Enzo, Farmingdale, NY), and mouse anti-NeuN (1/1000; Millipore). Fixed whole-mount SVZs were stained with mouse anti-acetylated tubulin (1/1000; Sigma), rabbit anti-β-catenin (1/500; Sigma), mouse anti-GFAP (1/3000; Sigma), and goat anti-EGFR (1/250; R&D Systems). Alexa Fluor 488-, 555-, and 647-conjugated secondary antibodies were used (Life Technologies, Carlsbad, CA).

## Western blotting

Cells were resuspended in 10% trichloracetic acid (TCA, Sigma). Extracts were incubated on ice for at least 15 min and centrifuged at 16,100×$g$ at 4°C for 15 min. Precipitates were washed in acetone twice and dried. The pellets were solubilized in 9M urea, 2% Triton X-100, and 1% DTT. LDS loading buffer (Invitrogen) was added and the pellet was heated at 70°C for 10 min. Samples were separated on Bis-Trispolyacrylamide gels (Invitrogen) and transferred to PVDF membrane (Millipore or BioRad, Hercules, CA). Membranes were treated with the SuperSignal Western Blot Enhancer (Thermo Scientific, Waltham, MA) and blots were developed with the SuperSignal West Femto chemiluminescence kit (Thermo Scientific). Blots were stripped with 1% SDS, 25 mM glycine (pH 2) prior to reprobing. The following primary antibodies were used for western blots: β-Actin (AC-15; Santa Cruz Biotechnology, Santa Cruz, CA), Bmi-1 (F6; Millipore).

## Quantitative PCR

Cells were sorted directly into RLT plus buffer (Qiagen, Venlo, Netherlands) supplemented with 2-mercaptoethanol. RNA was extracted with the RNeasy micro plus kit (Qiagen) and cDNA was synthesized with the RT² First Strand Kit (Qiagen). Reactions were run in 20 µl volumes with SYBR green and a LightCycler 480 (Roche Applied Science). Primer sequences were: Bmi-1 F (*Figures 6A*), 5′-CCAATGGCTCCAATGAAGACC-3′, Bmi-1 R (*Figures 6A*), 5′-TTGCTGCTGGGCATCGTAAG-3′, Bmi-1 F (*Figure 6—figure supplement 1G*), 5′-CGCTCTTTCCGGGGATCTTT-3′, Bmi-1 R (*Figure 6—figure supplement 1G*), 5′-CTCCACACAGGACACACATTA-3′, Cdkn2a (*p16^{Ink4a}*) F, 5′-GTGTGCATGAC GTGCGGG-3′, Cdkn2a (*p16^{Ink4a}*) R, 5′-GCAGTTCGAATCTGCACCGTAG-3′; β-actin F, 5′-CGTCGA CAACGGCTCCGGCATG-3′; β-actin R, 5′-GGGCCTCGTCACCCACATAGGAG-3′; Nestin F 5′-GGG CCCAGAGCTTTCCCACG-3′; Nestin R 5′-GGGCATGCACCAGACCCTGTG-3′.

## Technical considerations when sorting adult SVZ cells for neurosphere assays

Neurosphere assays were done as previously published by our lab (*Molofsky et al., 2003*, *2005*; *Nishino et al., 2008*). However when sorting adult SVZ cells for neurosphere assays several important technical points require attention:

## Debris elimination

The extensive debris in adult SVZ cell preps complicates the determination of which events are real cells. This is not a problem for embryonic or neonatal cell preparations. We compared multiple strategies to eliminate debris from adult SVZ preparations. The simplest, least expensive, and most reproducible solution was stringent forward scatter and side scatter gating as shown in *Figure 1—figure supplement 1A*. To use this technique it is important to calibrate the forward scatter and side scatter voltages using freshly isolated bone marrow cells because even a small change in PMT voltages can dramatically affect cell yield and the amount of debris contaminating the cellular fraction. Before every experiment we aligned the myeloid population into the center of the forward scatter/side scatter plot [FSC-Area = 125k, SSC-Area = 125k] then we ensured that the forward scatter threshold was set as high as possible without eliminating cells (typically 75000 and did not require daily adjustment). When the forward scatter and side scatter voltages are properly calibrated there should be a prominent population of adult SVZ cells with similar FSC/SSC as lymphoid cells.

There are drawbacks to stringent forward scatter and side scatter gating. Some debris has similar FSC/SSC characteristics as live SVZ cells (typically ~10% of events in the live SVZ scatter fraction represent debris as judged by DAPI staining after sorting on slides), and some cells are lost in the debris exclusion gate including a few cells with neurosphere-forming activity (generally ~25% of NICs are excluded by the debris exclusion gate) and all ependymal cells (which have very high side scatter properties due to their cilia). As a result we also explored other options to increase cell purity and yield which we will summarize here.

We tried two other techniques to eliminate debris: sucrose gradient and myelin depletion. For sucrose gradient, we resuspended pelleted SVZ cells in 10 ml of 0.9M sucrose in staining medium, then centrifuged the cells (750×*g*, 10 min, 4°C) and decanted the supernatant. The sucrose gradient typically eliminated ~90% of debris but also lost ~50% of cells. We performed myelin depletion using the Myelin Removal Beads II kit (130-096-733; Miltenyi Biotec) according to the manufacturer's instructions. Myelin depletion typically eliminated ~95–98% of debris but also eliminated ~50% of cells. Because of the unacceptable losses in cell yields, for routine experiments we did not eliminate debris using either of these methods. However when cell purity is paramount these debris elimination techniques may be useful.

We also attempted other techniques to enrich for cells and to deplete debris using vital dyes. We tried Calcein AM (C3099; Life Technologies) and CFSE (C1157; Life Technologies) but these stains did not distinguish adequately between cells and debris. We also tried Hoechst 33342 (H1399; Life Technologies) and Vybrant DyeCycle Violet (V35003; Life Technologies), and although these stains did distinguish between cells and debris we found that these reagents were not useful because (1) the intensity of staining varied substantially across days, (2) staining of other markers (in particular Glast) was negatively affected by using these reagents, and (3) efficiency of clonogenesis after sorting was negatively impacted by using either of these reagents. As a result we did not use these dyes for routine experiments, but they could be useful when cell yield is paramount.

Finally we also used the fluorescent transgenic lines *Nestin-GFP* and *Nestin-mCherry* (*Birbrair et al., 2011*; *Ding et al., 2012*). These lines allowed easy discrimination of SVZ cells from debris because of their bright, unambiguous expression in cells. Using *Nestin-GFP* or *Nestin-mCherry* and gating on GFP$^+$DAPI$^-$ (or mCherry$^+$DAPI$^-$) events typically gave ~50% greater cell yield and greater cell purity than when only using scatter-gating for cell identification. This is presumably because some Nestin-GFP+ or Nestin-Cherry+ cells (such as ependymal cells) appear as debris by FSC/SSC profile and would otherwise be excluded. However we did not wish to use transgenic lines to purify GEPCOT or pre-GEPCOT cells as this would limit the technique to these genetic backgrounds, and because express these transgenes are not uniformly expressed by pre-GEPCOT cells. However, for certain applications where high cell purity and yield are both paramount, or ependymal cells are desired, then *Nestin-GFP* or *Nestin-mCherry* may be useful.

## EDTA in staining medium increased GEPCOT cell yield

In our experiments the staining medium we employed contained 3.0 mM EDTA, enough to neutralize 2.7 mM of divalent cations in the staining medium (with a small surplus). We found that this slight excess of EDTA had a large effect (2–3-fold) on the yield of GEPCOT cells, probably by preventing clumping of GEPCOT cells with debris and/or other cells. The EDTA also prevented clogging of the

nozzle during sorting. EDTA was found to be better than EGTA for this purpose. However it was important to maintain the osmolarity of the staining medium when adding EDTA or clonogenesis was negatively impacted. We pre-diluted 0.5M EDTA (made from $Na_2H_2EDTA$, pH 8.0) to 77.7 mM (309 mOsm) before adding it to the staining medium.

## Growth medium pH must be kept neutral during sorting by preventing equilibration with the atmosphere

When sorting cells directly into culture medium it is essential to keep the pH of the medium constant. The phenol red indicator should maintain a red-orange color (pH 7.2–7.4) before, during, and after the sort. Typically we keep each plate in a Ziploc bag filled with 6% $CO_2$/93% $N_2$/1% $O_2$ prior to the sort, then sort one plate in 2–3 min or less, and return it to a freshly gassed Ziploc bag until transferring the plate to a humidified 37°C, $CO_2$ incubator for culture. By keeping plates in 6% $CO_2$ before and after the sort, the culture medium does not equilibrate with the air (which would deplete $CO_2$ from the medium and increase pH to levels incompatible with cell survival).

## Rock inhibitor (Y-27632) and IGF1 increase clonogenesis after sorting

The process of being sorted has a negative effect on neurosphere formation from adult SVZ cells: at the beginning of our experiments we observed a ~threefold loss of activity in sorted live SVZ cells compared to crude unsorted cells. Sorted neonatal SVZ cells do not show this problem. We wondered how to sort gently to overcome the fragility of adult SVZ cells.

Addition of the ROCK inhibitor, Y-27632 hydrochloride, to the culture medium after sorting (10 µM) increased neurosphere formation by adult cells approximately twofold after sorting but did not have any effect on adult SVZ cells pipetted into the culture medium, on sorted neonatal SVZ cells, or on crude neonatal SVZ cells. We also found that IGF1 (20 ng/ml) further improved clonogenesis after sorting. The mechanisms underlying these effects are unclear but may involve promotion of cell survival after sorting or repression of BMP signaling (*Hsieh et al., 2004*). When using both these culture additives, there was little difference between sorted live SVZ cells and an equal number of cells added to culture by pipette in terms of the numbers of neurospheres they made.

## Preventing neurosphere aggregation after sorting

Neurospheres can aggregate and fuse in culture, particularly when cultured at higher densities or when plates are moved (facilitating collisions among neurospheres) during the culture period (*Singec et al., 2006*). To minimize this problem we took several precautions. First, we always sorted cells into culture at clonal (low) density: 1000 cells in 1.5 ml medium (0.66 cells/µl) in each well of a 6 well-plate. Second, we maintained a separate incubator specifically for neurosphere cultures that was only opened sparingly to prevent mechanical disturbances or changes in $CO_2$ concentrations.

Finally, we found that chick embryonic extract (CEE) (*Molofsky et al., 2003*) can prevent cell fusion after sorting. In the absence of CEE, sorted cell-derived primary neurospheres were few in number and very large, suggesting extensive sphere fusion. After adding 10% CEE to the culture medium, sorted cell-derived primary neurospheres were indistinguishable in size and number from neurospheres cultured from cells pipetted into culture. We found that the combination of 10% CEE plus 10 µM Y-27632 hydrochloride plus 20 ng/ml IGF1 maximized the formation of non-aggregating neurospheres from sorted adult SVZ cells.

## Acknowledgements

SJM is a Howard Hughes Medical Institute Investigator and the Mary McDermott Cook Chair in Pediatric Genetics. This work was supported by the Ellison Medical Research Foundation, the National Institute on Aging (R37 AG024945) and the Cancer Prevention and Research Institute of Texas. RAJS was supported by fellowships from the Leukemia & Lymphoma Society (5541-11) and the Canadian Institutes of Health Research (MFE-106993). We thank G Enikolopov for *Tg(Nestin-GFP)* and *Tg(Nestin-mCherry)* mice, K Harter, and K Brummell for technical support, R Coolon, S Manning, M Gross and K Correll for managing the mouse colony, N Loof and the Moody Foundation Flow Cytometry Facility in CRI for flow cytometry, and BD Biosciences for antibodies.

# Additional information

## Competing interests

SJM: Reviewing editor, *eLife*. The other authors declare that no competing interests exist.

## Funding

| Funder | Grant reference number | Author |
| --- | --- | --- |
| Howard Hughes Medical Institute | None | Sean J Morrison |
| National Institutes of Health | R37 AG024945 | Sean J Morrison |
| Ellison Medical Research Foundation | Senior Scholar | Sean J Morrison |

The funder had no role in study design, data collection and interpretation, or the decision to submit the work for publication.

## Author contributions

JKM, RAJS, Conception and design, Acquisition of data, Analysis and interpretation of data, Drafting or revising the article; DN, JEJ, Conception and design, Contributed unpublished essential data or reagents; AP, RJB, Acquisition of data, Analysis and interpretation of data; TYV, Acquisition of data, Contributed unpublished essential data or reagents; SJM, Conception and design, Analysis and interpretation of data, Drafting or revising the article

## Ethics

Animal experimentation: This study was performed in strict accordance with the recommendations in the Guide for the Care and Use of Laboratory Animals of the National Institutes of Health. All of the animals were handled according to approved institutional animal care and use committee (IACUC) protocols (protocol# 2011-0104) of the University of Texas Southwestern Medical Center. Every effort was made to minimize suffering.

# Additional files

## Supplementary file

• Supplementary file 1. (**A**) List of antibody markers screened to purify neurosphere-initiating cells. (**B**) List of growth factors screened to stimulate pre-GEPCOT growth.

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
