## [Decision Letter]

Thank you for sending your work entitled “Prospective identification of functionally distinct neural stem cells and neurosphere-initiating cells in the forebrain” for consideration at *eLife*. Your article has been favorably evaluated by Janet Rossant (Senior editor) and 3 reviewers, one of whom is a member of our Board of Reviewing Editors.

The Reviewing editor and the other reviewers discussed their comments before we reached this decision, and the Reviewing editor has assembled the following comments to help you prepare a revised submission.

The general consensus of the reviewers was that the data are extensive, of high quality, use cutting edge approaches, and support the conclusions. In addition, the findings make a significant contribution to the field of adult neural stem cells, particularly with regard to the ongoing search for the “true” quiescent adult neural stem cell. However, the reviewers did raise a number of issues that, if addressed, would strengthen the manuscript, as follows:

1) The authors indicate that they could not grow colonies from the pre-GEPCOT cells under neurosphere conditions or under adherent conditions. Nonetheless, it is clear that these cells must proliferate under some conditions, since they repopulate the GEPCOT compartment following its ablation. Any data addressing potential growth factor requirements for these cells would greatly enhance the manuscript. If the authors have tried to identify such growth factors and have been unsuccessful then even that negative data would be informative.

2) Given the link between p16 and senescence, have the authors ever asked whether senescence is the relevant mechanism for depletion of neural stem cells in their various Bmi1 mouse models?

3) In Figure 6, there are many non-specific bands, and it is not clear whether Bmi-1 protein expression was lost in the mutant mice. The authors should improve the data, or alternatively perform qPCR to confirm the loss of Bmi-1 expression.

On a more minor note:

1) One of the surprising findings of this manuscript is the relatively high frequency of the pre-GEPCOT cells (6%). It is generally assumed in the field that the quiescent neural stem cells should be a rare population. The authors do indicate in the discussion that the pre-GEPCOT cells they are sorting are likely to be heterogeneous. However, it would be worth discussing this issue further.

2) It would be nice to see the enhanced GFAP staining that the authors allude to when they describe the floxed *Bmi1*;*Nestin-Cre* mice in Figure 6.

3) The proportions of new astrocytes and oligodendrocytes that are made from their BrdU-positive proliferating SVZ cells in vivo seem to be very small. It would help the reader if the authors were to compare the relative proportions of BrdU-positive cells that made neurons versus glia over the relevant timeframes.

---

## [Author Response]

Before getting into the reviewer comments I would like to note that in addition to the new data we have provided to address reviewer comments, we have also added other new data to strengthen the manuscript:

We have added new data extending the *GFAP-CreER*^*T2*^ lineage tracing to 35 days during regeneration after temozolomide (TMZ) treatment (Figure 5). In the original manuscript we had chased for only 22 days. This longer chase period strengthens the evidence that the cells that are responsible for the regeneration of GECPOT neurosphere-initiating cells (NICs) and other SVZ cells after TMZ ablation are GFAP^+^. Since GFAP is one of the most specific markers of pre-GEPCOT cells, this further supports our conclusion that quiescent pre-GEPCOT cells are the stem cells of the SVZ.

We have also added new data on neurogenesis in *Nestin-Cre; Bmi-1*^*fl/fl*^ mice. In the original manuscript we showed that *Bmi-1* deletion significantly reduced neurogenesis in 14-month-old *Nestin-Cre; Bmi-1*^*fl/fl*^ mice relative to littermate controls. In the revised manuscript we have expanded our analysis to include neurogenesis in 4-month-old *Nestin-Cre; Bmi-1*^*fl/fl*^ mice, showing that neurogenesis is normal in these mice (see Figure 6). This further demonstrates that even when *Bmi-1* is deleted throughout the nervous system during fetal development, stem cells and neurogenesis persist into young adulthood and are not depleted until around a year of age. Bmi-1 is therefore not absolutely required for the maintenance of adult neural stem cells, in contrast to what has been widely assumed based on studies of germline knockout mice.

Review comments:

*1) The authors indicate that they could not grow colonies from the pre-GEPCOT cells under neurosphere conditions or under adherent conditions. Nonetheless, it is clear that these cells must proliferate under some conditions, since they repopulate the GEPCOT compartment following its ablation. Any data addressing potential growth factor requirements for these cells would greatly enhance the manuscript. If the authors have tried to identify such growth factors and have been unsuccessful then even that negative data would be informative*.

We agree that it should, in principle, be possible to identify culture conditions that enable colony-formation by pre-GEPCOT cells and have worked very hard to do this. To date, we have tested the effects of 36 different growth factors or medium supplements on adherent or non-adherent colony formation by SVZ cells from normal or TMZ-treated mice. Since TMZ treatment ablates GEPCOT NICs, any growth factor that enables colony formation by pre-GEPCOT cells should profoundly increase colony formation by SVZ cells from TMZ-treated mice. Out of the 36 growth factors and medium supplements we have tested so far, 5 significantly reduced colony-formation but none significantly increased colony formation. The new data have been added to Supplementary File 1B. We will continue testing additional growth factors and medium supplements in future studies.

*2) Given the link between p16 and senescence, have the authors ever asked whether senescence is the relevant mechanism for depletion of neural stem*
*cells in their various Bmi1 mouse models?*

We have added new data further exploring *p16*^*Ink4a*^ expression after *Bmi-1* deletion. In the original manuscript we demonstrated in *Nestin-Cre; Bmi-1*^*fl/fl*^ mice that *p16*^*Ink4a*^ expression increased after *Bmi-1* deletion in pre-GEPCOT but not GEPCOT cells (see Figure 6). In the revised manuscript, we have added new data demonstrating in *Nestin-CreER*^*T2*^*; Bmi-1*^*fl/fl*^ mice that *p16*^*Ink4a*^ expression also increased in pre-GEPCOT but not GEPCOT cells (see Figure 7). These results demonstrate that quiescent neural stem cells are particularly dependent upon Bmi-1 for the repression of *p16*^*Ink4a*^ expression, providing a molecular distinction in the way in which Bmi-1 functions in pre-GEPCOT versus GEPCOT cells.

The most commonly used marker of cellular senescence is β-galactosidase staining. We have attempted to test whether *Bmi-1* deletion induced cellular senescence by staining for senescence-associated β-galactosidase activity. However, relative to control cells, we have not been able to detect any increase in senescence-associated β-galactosidase activity in pre-GEPCOT cells from 14-month-old *Nestin-Cre; Bmi-1*^*fl/fl*^ mice or in cultured neural stem/progenitor cells from *Bmi-1* germline knockout mice. We have added this information to the text of the revised manuscript; however, we cannot entirely exclude the possibility of senescence among *Bmi-1* deficient neural stem cells as there are no definitive markers of cellular senescence in vivo.

We are also attempting to test whether *p16*^*Ink4a*^ deficiency rescues the depletion of pre-GEPCOT and GEPCOT cells after *Bmi-1* deletion but have not yet been able to generate definitive results for technical reasons. We initially attempted to test this by generating *Nestin-CreER*^*T2*^*; Bmi1*^*fl/fl*^*; p16*^*Ink4a*^*-p19*^*Arf -/-*^ compound mutant mice. However, germline deletion of *p16*^*Ink4a*^*-p19*^*Arf*^ leads to the death of mice from hematopoietic malignancies by around 5 months of age. Therefore, these mice died before we could test whether there was a rescue of neural stem cell function. We have more recently generated *Nestin-CreER*^*T2*^*; Bmi1*^*fl/fl*^*; p16*^*Ink4a fl/fl*^ mice but will have to age these mice for at least a year to test whether conditional *p16*^*Ink4a*^ deletion rescues the depletion of *Bmi-1* deficient pre-GEPCOT and GEPCOT cells (see Figure 7).

*3) In*
Figure 6*, there are many non-specific bands, and it is not clear whether Bmi-1 protein expression was lost in the mutant mice. The authors should improve the data, or alternatively perform qPCR to confirm the loss of Bmi-1 expression*.

We have redone the western blot in Figure 6 and have replaced the blot shown in the original manuscript with another blot that is cleaner and includes more controls. Consistent with our conclusions in the original manuscript, the new blot shows that conditional deletion of *Bmi-1* using *Nestin-Cre* leads to a complete or near complete, loss of Bmi-1 protein throughout the cortex and SVZ (see new Figure 6). We have also included new RT-PCR data demonstrating that SVZ cells, GEPCOT cells, and pre-GEPCOT cells from *Nestin-Cre; Bmi-1*^*fl/fl*^ mice generally lack detectable *Bmi-1* mRNA, in contrast to littermate controls (see new Figure 6—figure supplement 1). These new data supplement the genomic DNA PCR data in the original manuscript showing that SVZ cells, GEPCOT cells, and pre-GEPCOT cells from *Nestin-Cre; Bmi-1*^*fl/fl*^ mice all exhibit complete *Bmi-1* recombination (see Figure 6). Thus, PCR analysis of genomic DNA, RT-PCR analysis of mRNA, and western analysis of protein all indicate efficient deletion of *Bmi-1* from SVZ cells, GEPCOT cells, and pre-GEPCOT cells.

*On a more*
*minor note:*

*1) One of the surprising findings of this manuscript is the relatively high frequency of the pre-GEPCOT cells (6%). It is generally assumed in the field that the quiescent neural stem cells should be a rare population. The authors do indicate in the discussion that the pre-GEPCOT cells they are sorting are likely to be heterogeneous. However, it would be worth discussing this issue further*.

We have added new text to discuss this issue. There are two important points. First, the pre-GEPCOT cell population is probably not pure. However, given the lack of a clonal assay to detect quiescent neural stem cells we have no way to test purity. This means that to exclude non-stem cells from this population we will have to perform fate mapping with each potential non-stem cell marker. This will take a long time. Second, we should not expect quiescent neural stem cells to be rare. The idea that quiescent stem cells should be rare comes from the hematopoietic system, where stem cells are always rare and are always fated to form massive numbers of progeny. The nervous system has adopted a fundamentally different strategy in which stem cells (when they are present) are common, but are each fated to form relatively small numbers of progeny in vivo. Fate mapping of neural stem cells in multiple regions of the developing and adult nervous system has shown that undifferentiated neural stem cells undergo only a small number of divisions before terminally differentiating (e.g. PNAS 110: E1045; Cell Stem Cell 8: 566; Cell 145: 1142). Neural stem cells are commonly estimated to account for a few percent of the cells in germinal zones of the adult nervous system (e.g., Cell Stem Cell 11: 265; PNAS 106: 6387; J. Neurosci. 17: 5046).

*2) It would be nice to see the enhanced GFAP staining that the authors allude to when they describe the floxed* Bmi1;Nestin-Cre *mice in*
Figure 6.

We have added a new figure showing the dramatic increase in GFAP staining in the forebrain of young adult *Nestin-Cre; Bmi-1*^*fl/fl*^ mice (see Figure 6—figure supplement 1). This effect was not observed in young adult *Nestin-CreER*^*T2*^*; Bmi-1*^*fl/fl*^ mice. This appears to be a developmental consequence of *Bmi-1* deficiency in neural stem/progenitor cells or in glia.

*3) The proportions of new astrocytes and oligodendrocytes that are made from their BrdU-positive proliferating SVZ cells in vivo seem to be very small. It would help the reader if the authors were to compare the relative proportions of BrdU-positive cells that made neurons versus glia over the relevant timeframes*.

We have revised the text to make clear that the vast majority of newborn cells that arise in the adult olfactory bulb are neurons, such that the proportion of new astrocytes and oligodendrocytes in our experiments was very small. This is consistent with a number of prior studies. Lineage tracing by multiple techniques (Cre/Lox, retrovirus, and BrdU) indicates that olfactory bulb neurons make up the vast majority of SVZ-derived mature cells in the mouse brain (e.g., Science 317: 381; J. Neurosci. 27: 12623; Nature 443: 448). Newborn mature glia are generated, but on a much smaller scale (e.g., J. Neurosci. 26: 7907; Genesis 47: 122).